# PEAC-seq adopts Prime Editor to detect CRISPR off-target and DNA translocation

Zhenxing Yu[1,2,3,4,7], Zhike Lu[2,3,4,7], Jingjing Li[3,5,7], Yingying Wang[1,2,3,4,7], Panfeng Wu[3,6], Yini Li[1,2,3,4], Yangfan Zhou [2,3,4], Bailun Li[2,3,4], Heng Zhang[3,4], Yingzheng Liu[2,3,4] & Lijia Ma [2,3,4] ✉

CRISPR technology holds significant promise for biological studies and gene therapies because of its high flexibility and efficiency when applied in mammalian cells. But endonuclease (*e.g.*, Cas9) potentially generates undesired edits; thus, there is an urgent need to comprehensively identify off-target sites so that the genotoxicities can be accurately assessed. To date, it is still challenging to streamline the entire process to specifically label and efficiently enrich the cleavage sites from unknown genomic locations. Here we develop PEAC-seq, in which we adopt the Prime Editor to insert a sequence-optimized tag to the editing sites and enrich the tagged regions with site-specific primers for high throughput sequencing. Moreover, we demonstrate that PEAC-seq could identify DNA translocations, which are more genotoxic but usually overlooked by other off-target detection methods. As PEAC-seq does not rely on exogenous oligodeoxynucleotides to label the editing site, we also conduct in vivo off-target identification as proof of concept. In summary, PEAC-seq provides a comprehensive and streamlined strategy to identify CRISPR off-targeting sites in vitro and in vivo, as well as DNA translocation events. This technique further diversified the toolkit to evaluate the genotoxicity of CRISPR applications in research and clinics.

CRISPR-based genome editing exhibited enormous potential in both biological research and clinical applications. Compared to small-molecule drugs and antibody drugs, CRISPR therapy has the unique advantage of directly targeting the nucleic acid sequences of previously undruggable targets. However, non-specific targeting of gRNAs, which might introduce undesired edits, causes unexpected cell genotoxicity. And it is urged to understand the outcomes of off-target edits and the resulting DNA translocations, which challenges the great translational potential of CRISPR technology in harnessing genetic disorders and other human diseases.

To date, versatile tools have been developed to identify CRISPR off-target sites. In vitro techniques capture nuclease-induced cleavage events directly from purified genomic DNA or chromatin[1,2]; these approaches typically require 400–500 million reads per sample to identify off-targets. Some other methods incorporated enrichment of fragmentized DNA by circularizing sequences[3] or by introducing biotinylated oligos to overcome the high sequencing requirement[4]. However, in vitro techniques typically reported many sites that did not occur in a cellular context, and methods for *in cellula* and in vivo detection are highly demanded. GUIDE-seq labeled and enriched

[1]Fudan University, 220 Handan Road, 201100 Shanghai, China. [2]Center for Genome Editing, Westlake Laboratory of Life Sciences and Biomedicine, 18 Shilongshan Road, 310024 Hangzhou, Zhejiang, China. [3]School of Life Sciences, Westlake University, 600 Dunyu Road, 310030 Hangzhou, Zhejiang, China. [4]Institute of Biology, Westlake Institute for Advanced Study, 18 Shilongshan Road, 310024 Hangzhou, Zhejiang, China. [5]Reproductive Medicine Center, Tongji Hospital, Tongji Medical College, Huazhong University of Science and Technology, 430030 Wuhan, Hubei, China. [6]Department of Tissue and Embryology, School of Basic Medical Sciences, Wuhan University, 115 Donghu Road, 430071 Wuhan, Hubei, China. [7]These authors contributed equally: Zhenxing Yu, Zhike Lu, Jingjing Li, Yingying Wang. ✉e-mail: malijia@westlake.edu.cn

double-strand breaks (DSBs) in the genome of living cells using exogenous double-stranded oligodeoxynucleotides (dsODNs) mediated by an end-joining process[5]. However, the high molarity of exogenous dsODNs limited its application to detect off-targets for in vivo CRISPR editing. BLISS is another type of *in cellula* technique, which utilizes in situ DSB ligation in fixed cells and characterizes the off-target sites for both SpCas9 and As/LbCpf1[6]. As CRISPR technology holds therapeutic potential for many unmet medical needs, the off-target identification of in vivo CRISPR editing and the evaluation of corresponding genotoxicity are highly demanded. To do so, one strategy is to use in vitro or computational approaches to prioritize a list of genomic regions and validate them on in vivo samples one by one through targeted amplicon sequencing (Amplicon-seq)[7–9], which risked overlooking in vivo specific off-targets and suffered from tedious labor work if the prior data comes with a long candidate list. DISCOVER-seq, however, utilized the signal of chromatin immunoprecipitation of MRE11, which is involved in the DNA repairing pathway, to represent and enrich genomic sites undergoing DSB-induced repairs[10]. However, the dynamic nuclease activity of Cas9 might not be fully captured by the "snapshot" signal from MRE11 immunoprecipitation.

Further, DNA translocation has been a significant concern for CRISPR editing, as it typically causes higher genotoxicity, although it occurs at a relatively lower frequency[11]. The potential risk of DNA translocation has often been concentrated on applying CRISPR editing in producing CAR-T cells since multiple gRNAs were introduced to T cells and cause risks of translocation between double-strand DNA (DSB) ends[12,13]. Methods have been developed to identify DNA translocations, but the sequence information of at least one end of the rearranged DNAs is usually required, e.g., HTGTS[14–17]. And a systematic identification of DNA translocation is still lacking.

Here, we introduce an off-target identification method, PEAC-seq (Prime Editor Assisted off-target Characterization), in which we design a Cas9-MMLV fusion protein to take advantage of the sequence insertion ability from the Prime Editor (PE)[18]. The native PE system (Cas9n-MMLV) utilizes a pegRNA (Prime Editor gRNA) containing extra sequences at the 3' of gRNA, which serve as a priming site and allow reverse transcription (RT) from the exposed 3'-hydroxyl group of the non-targeting strand to incorporate additional DNA sequences into the editing sites. In PEAC-seq, an optimized RT template is used to incorporate PEAC-seq tag sequences, which are further used to represent and enrich the local sequences of the edited sites from the genome, including both on-target and off-target sites. PEAC-seq accompanies the process of CRISPR editing and tag insertion, which ensures the consistency between editing events and PEAC-seq signals. We apply PEAC-seq on a few promiscuous sites in both *in cellula* and in vivo samples and demonstrate that PEAC-seq could effectively identify off-targets by comparing to the results of GUIDE-seq, DISCOVER-seq, WGS, and Amplicon-seq. Furthermore, benefiting from the directional inserted PEAC-seq tag, we successfully identify DNA translocations, which could not be directly profiled by current methods and are typically more toxic to cells. Together, PEAC-seq is an unbiased method of identifying CRISPR off-targets and off-target-related DNA translocations. As it bypassed the addition of high molarity of exogenous dsODNs, PEAC-seq also holds immense potential to identify off-targets and translocations for in vivo CRISPR editing, which would be particularly valuable for translational studies.

## Results

### Develop PEAC-seq for unbiased identification of CRISPR off-targets

To be compatible with off-target detection of in vivo CRISPR editing, we reasoned that the detection method should streamline the editing and off-target enrichment processes without relying on exogenous moiety. To do so, we adopted the prime editor system using Cas9 instead of Cas9n and utilized pegRNA to be templated for inserting a tag sequence for enrichment. The Cas9/pegRNA creates double-strand breaks (DSBs) in the genome at both on-target and off-target sites, and the tag sequence will be introduced at the DSB sites through reverse transcription from the pegRNA and incorporated into the genome through DNA repair. We designed a 21-nt insertion tag, with the consideration of (1) avoiding the RNA secondary structure of the insertion tag and between the insertion tag and the gRNA scaffold; (2) sequence uniqueness to the host genome; (3) sufficiently long for efficient anneal by PCR primers for enrichment. We named this assay PEAC-seq, Prime Editor Assisted off-target Characterization, as it employed the insertion ability of the Prime Editor to label and enrich the editing sites.

To enrich the genomic regions embedded with the PEAC-seq tag sequences, we adopted a priming strategy as GUIDE-seq[5]. We used Tn5 tagmentation instead of sonication to streamline the workflow and lower the starting DNA requirement (Fig. 1a). The UMI-included adapters were embedded into Tn5 to enable the elimination of PCR duplications from the sequencing data. During the library preparation, one of the biggest challenges is to effectively enrich the inserted tag sequences, whose length might vary. Since the PEAC-seq tag was from reverse transcription and extended alongside the RT template, both partial and full-length products might exist. Hence, primers must be carefully designed to enrich the editing sites with insertion at different lengths. To optimize the enrichment, we designed forward and reverse primers with different lengths of annealed base pairs to the inserted tag and evaluated their performances. We used three forward primers and two reverse primers with different extension starting points on the PEAC-seq tag (Fig. 1b). Different amplicons were generated in five separate reactions, each reaction was amplified by a forward primer and downstream Tn5 primer, or an upstream Tn5 primer and a reverse primer (Supplementary Fig 1). The enrichment to the PEAC-seq tag was evaluated to choose the best-performed primer set. It is worth pointing out that all the primers were designed at least 2-bp away from the insertion boundary so that the extension sequence could be used to filter out random priming reads (Fig. 1b).

Next, we examined the indel efficiency of PEAC-seq and the insertion efficiency of PEAC-seq tag at ten on-target sites. Across the ten examined sites, the indel frequencies of Cas9-MMLV and Cas9 are highly consistent (Supplementary Fig. 2a). And the insertion efficiencies of the full-length tag were 11–31% (Supplementary Fig. 2b–h), which is comparable to GUIDE-seq[5]. Encouraged by these pilot data, we conducted PEAC-seq in HEK293T cells at six sites (*VEGFA TS1*, *VEGFA TS2*, *VEGFA TS3*, *EMX1*, *FANCF*, and *RNF2*) that have been tested in multiple studies[1–5]. We used a modified GUIDE-seq analysis pipeline to rank and filter the identified editing sites. We analyzed the off-target sites generated from different primer sets for PEAC-seq tag enrichment and chose the F1 and R2 primers as the enrichment primers in the following analysis (Tables S1–S6, Supplementary Data 1–6, Supplementary Figs. 3–8, and "Methods−PEAC-seq in HEK293T cell").

At the sites of *VEGFA TS1*, *VEGFA TS2*, and *VEGFA TS3*, a large proportion of PEAC-seq off-targets were also reported by GUIDE-seq, but both methods hold a few unique off-targets (Fig. 1c and Supplementary Figs. 4 and 5). At the sites of *FANCF*, *EMX1*, and *RNF2*, all PEAC-seq off-targets were reported by GUIDE-seq (Fig. 1d, e and Supplementary Fig. 7). We then conducted Amplicon-seq to verify those off-targets that were only identified by GUIDE-seq or PEAC-seq at *VEGFA TS1*, *FANCF*, and *EMX1* sites[5]. At the *VEGFA TS1* site, Amplicon-seq confirmed the two PEAC-seq-unique off-targets, demonstrating good sensitivity of PEAC-seq. For the GUIDE-seq-unique off-targets, all six off-targets at the *FANCF* site were confirmed not to occur in our sample, while two out of the twelve GUIDE-seq-unique off-targets at the *EMX1* site and two out of the eight GUIDE-seq-unique off-targets at the *VEGFA TS1* site were detected by Amplicon-seq. These data argued that

PEAC-seq could effectively and specifically identify off-targets with a streamlined procedure without incorporating other exogenous reagents to tag and enrich these sites.

Next, we looked up the PEAC score and local sequences at the shared and unique off-targets. The PEAC score, calculated from the sequencing reads of PEAC-seq, quantitatively represents the enrichment of PEAC-seq tag at the edited sites. At *VEGFA TS1*, the off-target sites identified by both PEAC-seq and GUIDE-seq show higher PEAC scores compared to PEAC-seq-unique off-targets (Fig. 2a). Further, the number of sequencing reads surrounding the off-targets were highly correlated at the fourteen shared sites (Fig. 2b), suggesting their

consistency in detecting high confident off-targets. Noticeably, when examining the signal tracks of the PEAC-seq reads, the on-target site, shared off-target sites, and PEAC-seq-unique off-target sites show similar tracks (Fig. 2c). Also consistent with previous reports, the shared off-target sites composed a smaller number of mismatches than off-target sites unique to one of the methods (Fig. 2d), which is expected as the number of mismatches closely relates to the occurrence of off-target editing.

Furthermore, we also examined whether the position of mismatches on the pegRNA sequence might affect the off-target identification[5], especially in the primer binding site (PBS) that is

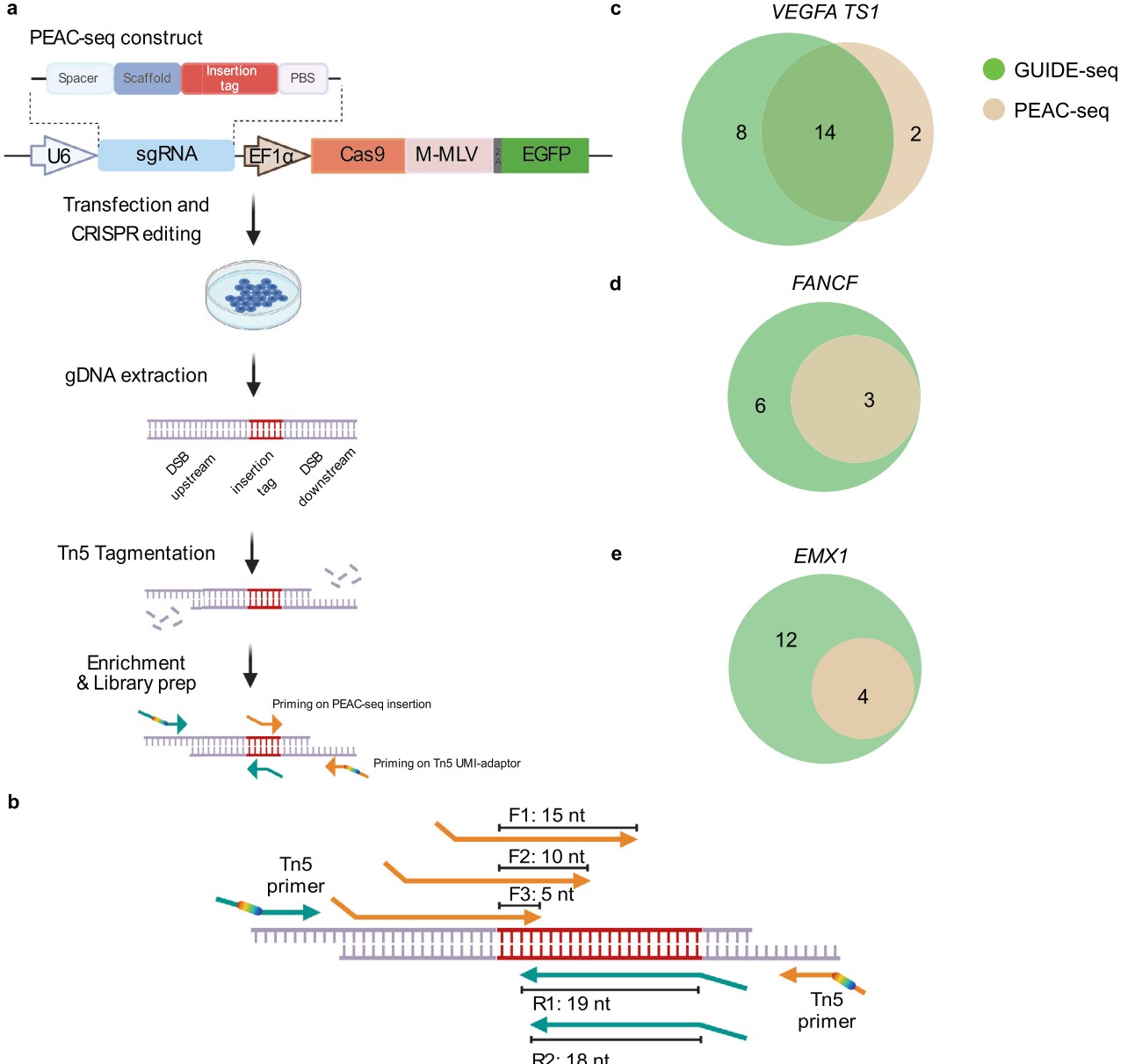

**Fig. 1 | Development of the PEAC-seq technique. a** Schematic representation of the PEAC-seq experimental procedure. The gDNA was extracted and undergone Tn5 tagmentation. The Tn5 was embedded with UMI adapters to eliminate PCR duplications in silicon. After tagmentation, fragments were amplified by pairs of primers (one priming at the PEAC-seq insertion, another priming with the Tn5 adapter). **b** Schematic representation of the three forward primers and two reverse primers designed for tag enrichment and library preparation of PEAC-seq. Each forward primer was paired with a downstream Tn5 primer to generate amplicons including the PEAC-seq tag sequence and its downstream genomic sequences. Each

reverse primer was paired with an upstream Tn5 primer to generate amplicons including the PEAC-seq tag sequence and its upstream genomic sequences. In total, five Amplicon-seq data from the three forward primers and two reverse primers were generated, and six candidate lists of putative off-targets were inferred from the five Amplicon-seq data using a modified GUIDE-seq analysis pipeline ("Methods"). **c**–**e** Venn diagram shows the shared and unique off-targets identified by PEAC-seq and GUIDE-seq. The *VEGFA TS1* (**c**), *FANCF* (**d**), and *EMX1* (**e**). Source data are provided as a Source data file.

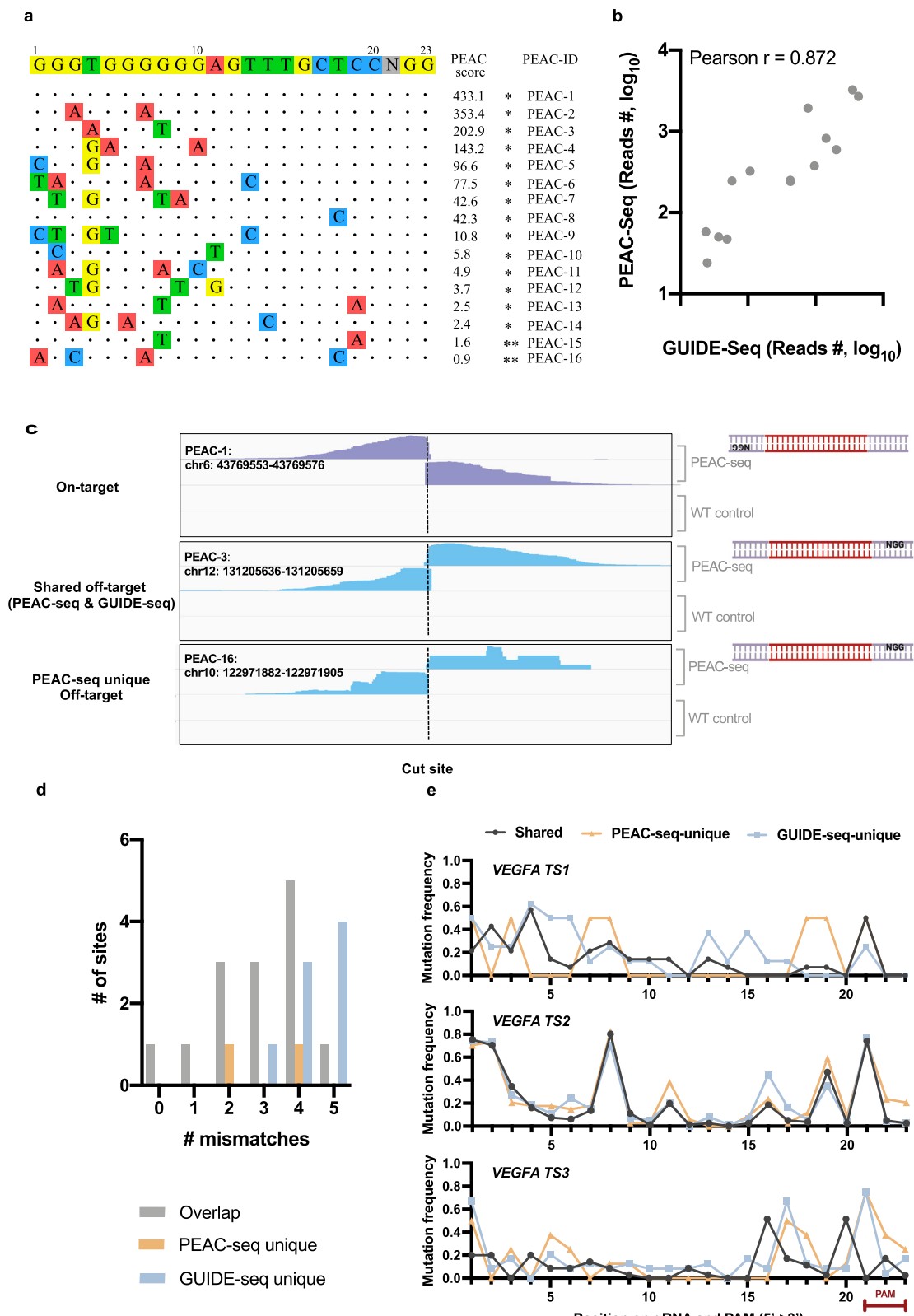

crucial to initiate the primer extension of reverse transcription[18]. To do that, we grouped the off-target sequences from the "Shared," "PEAC-seq-unique," and "GUIDE-seq-unique" and aligned with the on-target sequence and PAM sequences. The frequency at each position were plotted for the three sites (Fig. 2e). The patterns among the shared and unique off-target groups were quite consistent in *VEGFA TS2* (81 sites) and *VEGFA TS3* (35 sites), but a bit fluctuated in *VEGFA TS1* (24 sites).

Although the smaller number of off-targets of *VEGFA TS1* might contribute to its fluctuated mutation frequency, this result indicated that the sensitivity of PEAC-seq might be affected by mismatches located in the PBS region of PEAC-seq. Actually, the two verified GUIDE-seq-unique off-targets of TS1 both show mismatches in PBS region (at the position 14 and 17 of the spacer, respectively) (Table S10). Nevertheless, off-target identification of the TS3 gRNA seems more tolerant

**Fig. 2 | Analysis of the PEAC-seq off-target sites. a** The visualization of PEAC-seq on-target and off-target sites. The '*' represented a PEAC-seq site that was also called by the GUIDE-seq. The '**' represented a PEAC-seq off-target (PEAC-seq-unique) that was identified by Amplicon-seq but not called by the GUIDE-seq. PEAC score: quantitative enrichment of the PEAC-seq tag at the edited sites; PEAC-ID: each identified site (on-target and off-target) by PEAC-seq were assigned a PEAC-ID, which was ordered by the PEAC score (descending order). **b** The number of reads from the shared PEAC-seq and GUIDE-seq sites is highly correlated. **c** Screenshots of PEAC-seq signal tracks from the IGV Genome Browser. One on-target site, one shared off-target site, and one PEAC-seq unique off-target site were presented. For each site, signals from both the PEAC-seq and the wild-type (WT, no Cas9-MMLV

treatment) samples were included. For each sample, the first track represented signals from the amplicons of a forward primer and a downstream Tn5 primer; the second track represented signals from the amplicons of a reverse primer and an upstream Tn5 primer. The model on the right side showed the direction of the spacer and PAM of each case. **d** The shared off-targets (gray bars) tend to have less mismatches compared to the on-target site, while the PEAC-seq unique sites (orange bars) and the GUIDE-seq unique sites (blue bars) tend to have more mismatches. **e.** Mutation frequencies were plotted at each position alongside the gRNA and PAM sequences (from 5' to 3'). From top to bottom are profiles of *VEGFA TS1*, *TS2*, and *TS3*. Source data are provided as a Source data file.

to PBS mutations, which implied that the extent of the influence might be site-specific.

## PEAC-seq identified Cas9-dependent chromosome rearrangement

To enrich PEAC-seq tag, the forward primer (F1) and downstream Tn5 primer (R1) would amplify regions downstream, but not upstream, of the PEAC-seq tag (Fig. 3a). Surprisingly, in some cases, we saw unexpected signals located at the upstream genomic region of the F1-R1 amplicons (Fig. 3a and Supplementary Fig. 9). With further analysis on these sites, we speculated that the signals might come from the joining of DSB ends from another genome breaking site. As shown in the proposed models (Fig. 3b), PEAC-seq generates DSBs with three different ends, including one upstream end appended with a complete or partial PEAC-seq tag (②), one upstream end without PEAC-seq tag (①), and one downstream end (③). If multiple DSBs simultaneously occurred in nucleus and physically proximal to each other, DSB ends from different breaking points might join together and cause DNA rearrangements. In our hypothesized scenario, the upstream end with the PEAC-seq tag from a distal Donor Site may join with the upstream end of a Receiver Site, but the direction of the PEAC-seq tag is reverse relative to the Receiver Site (Fig. 3b, model (v)). This joining generates signals upstream to the PEAC-seq tag of the Receiver Site, which won't be amplified by the F1 and Tn5 primers (R1) (Fig. 3a).

The DSB-induced DNA rearrangements, which have not been systematically evaluated by other CRISPR off-target identification techniques, would cause severe chromosome aberrant including large fragment deletion, inversion, and translocation. Benefited from the directional PEAC-seq tag insertion, the resulting PCR amplicons could be used as indicators for chromosome rearrangements, as it could distinguish whether the amplicon came from the joining of expected DSB ends. To test this, we designed primers (Nest-F) located upstream of the F1 primer, which paired with the downstream Tn5 primer to identify the sequences of the unknown Donor sites (Fig. 3c, "Methods−Translocation characterization"). Noteworthy, a successful amplification bridging the Donor and the Receiver sites do not require the existence of the PEAC-seq tag insertion (Fig. 3b, models (III) and (iv)), which allowed us to comprehensively estimate the various rearrangement patterns between the Donor and the Receiver sites.

We conducted the Unidirectional Targeted Sequencing (UDiTaS)[17] at two susceptible sites and identified three types of translocations (Fig. 3d). The results indicated that both the upstream end (Fig. 3d, model (iii)) and the downstream end (Fig. 3d, model (iv)) of a distal Donor site could join with the upstream end of a Receiver site. This joining could happen either with or without the PEAC-seq insertion. And also, we identified many other translocations, some of which were between the selected sites and other on- and off-target sites, or other DSB sites (Supplementary Fig. 9b, c).

Interestingly, the frequencies of DNA translocation varied across different sites (Fig. 3e), and it did not necessarily happen between DSB ends with high indel frequencies. For example, among the PEAC-seq off-targets of *VEGFA TS3* sites, the on-target site (chr6:

43769716−43769739) shows a 0.2% translocation rate in our data, while at another off-target site (chr22: 37266776−37266799), 34.7% edits involve DNA translocations (Supplementary Fig. 9a). The translocation score of other *VEGFA TS3* off-targets and the other seven sites were provided (Supplementary Fig. 9a, Tables S1−8, Supplementary Data 1−8). These results suggested that the PEAC-seq could successfully identify chromosome translocations, further enabling the safety evaluation of the CRISPR application.

## Apply PEAC-seq for in vivo off-target detection

PEAC-seq used the templated information on pegRNA to insert tag sequences and not rely on exogenous tags. This straightforward procedure allowed us to investigate its application in vivo. We edited mice embryos at the pronuclear stage by injecting in vitro transcribed Cas9-MMLV mRNA and pegRNAs targeting *PCSK9* and *PNPLA3*. We collected embryos around E14.5 to E21 and generated the PEAC-seq off-target lists for these two sites (Fig. 4). We identified one *PCSK9* on-target and one off-target from the two embryos, which both have been previously reported by DISCOVER-seq[10] (Fig. 4b−d, Table S7, and Supplementary Data 7). Amplicon-seq verified the edits at the PEAC-seq off-targets and confirmed non-edits at the other reported off-targets. The small number of *PCSK9* off-targets in our study might be relevant to the short editing time window by using mRNA injection in embryos, compared to the adenovirus delivery in the liver[10]. Using the same strategy, we also conducted the PEAC-seq at another in vivo CRISPR therapy target *PNPLA3*. Three editing sites, including the on-target site, were identified by PEAC-seq from two embryos (Supplementary Fig. 10, Table S8, and Supplementary Data 8). These off-targets were also reported by previous in vivo study[10,19] and verified by Amplicon-seq. These data demonstrated the potential of PEAC-seq to directly detect off-targets in vivo, although more editing systems need to investigate.

## ePEAC-seq, an improved version of PEAC-seq utilizing epegRNA

Since the original PEAC-seq protocol has been developed, multiple strategies have been proposed to improve the editing efficiency of the native PE system, including modifications on pegRNA[20], MMLV[21], and transient expression of a dominant negative MMR (DNA mismatch repair) protein[22]. By incorporating epegRNA (engineered pegRNA, incorporated 3' RNA structural motif evopreQ$_1$), hMLH1, and epegRNA plus MLH1dn, we developed three modified versions of PEAC-seq and benchmarked their performances on identifying off-targets at *EMX1* and *VEGFA TS2* sites (Fig. 5a). We did not include the truncated MMLV, as it is reported to be effective in plants but not in mammal cells[21]. We specifically concentrated on the PEAC-seq tag insertion, whose efficiency is critical to the overall performance of PEAC-seq. Among the three modifications, incorporating epegRNA appears to be the most effective one to increase the number of PEAC-seq tag insertion at different cutoffs (Fig. 5b). We named the epegRNA version of PEAC-seq as ePEAC-seq. Importantly, ePEAC-seq successfully identified the two missed off-targets of *EMX1* (Fig. 5c, d), emphasizing its higher sensitivity than PEAC-seq. At the *VEGFA TS2* site, ePEAC-seq also called more off-target sites shared with GUIDE-seq, compared to PEAC-seq (Supplementary Figs. 4a and 11).

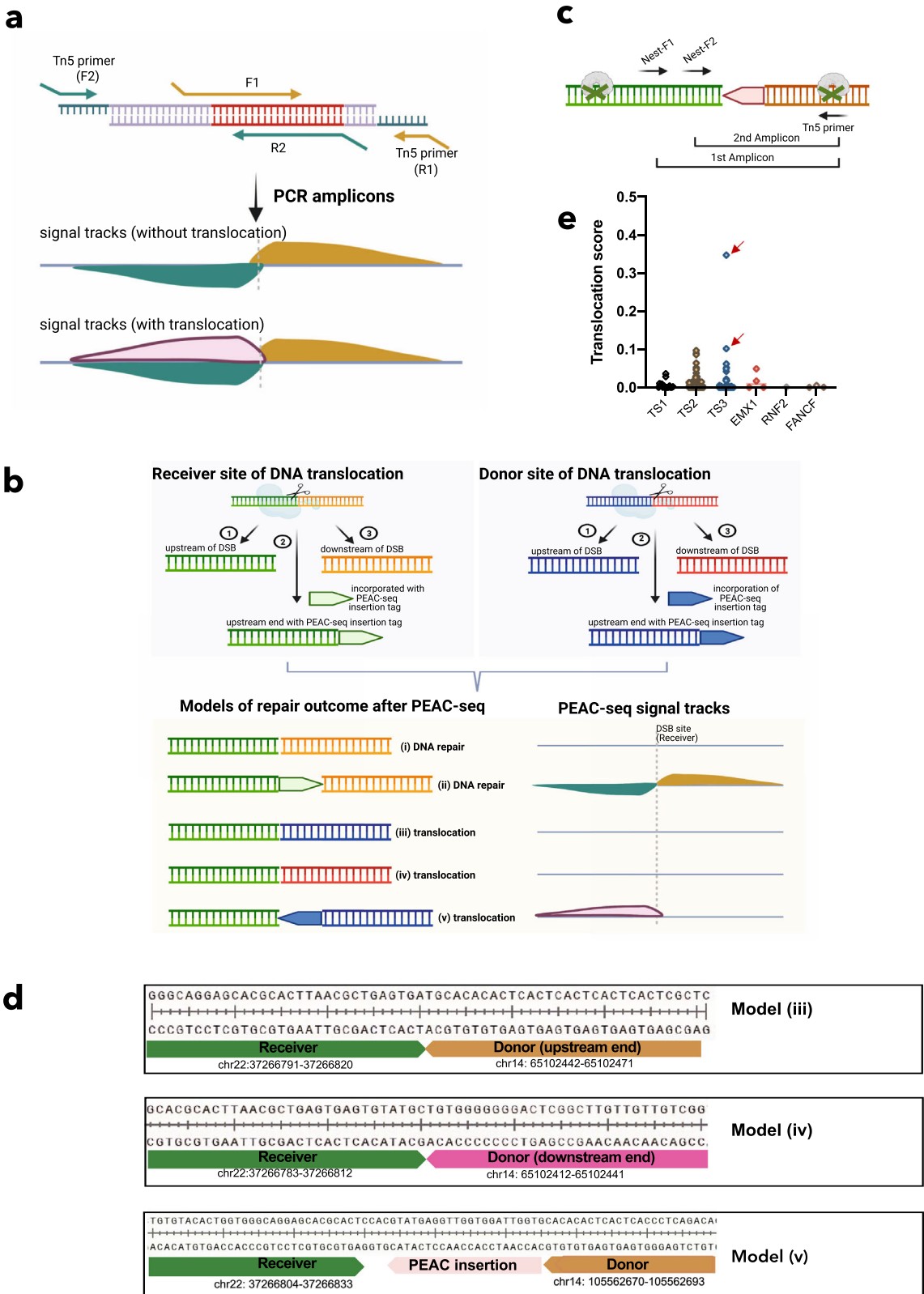

## Discussion

The off-target detection is crucial to the biotechnological and clinical applications of CRISPR technology. Over the past years, many elegant designs have been applied to depict the profile of off-targets in vitro and *in cellula*. These methods could label and enrich the cleavage sites without knowing the genomic locations of off-targets, but the addition of exogenous dsODN or chemicals limits their applications in vivo. Besides these experimental approaches, computational algorithms considered the diverse features of gRNA also contributed to generate a candidate off-target list. However, it is always a concern how well the cellular context could be reflected by these alternative approaches. To bypass the addition of exogenous agents, we adopted the Prime Editor to insert a tag sequence along with the cleavages. We used Cas9 instead of Cas9n to fuse

**Fig. 3 | PEAC-seq identified DNA translocations relevant to CRISPR genome editing. a** Signal tracks of one PEAC-seq site with unexpected upstream signals from the F-primer amplicon. Dashed gray bar: cutting site; Earthy yellow peak: expected signals from the F-primer; Pink peak: unexpected signals from the F-primer. **b** Proposed models of the generation of unexpected upstream signals. Both the Receiver site and the Donor site could generate DSBs and proximal to each other within the nucleus. Models (i) and (ii) joined DSB ends from the same Receiver site. Models (iii), (iv), and (v) joined one donor DSB and one Receiver DSB. If the donor DSB carried the PEAC-seq insertion, the unexpected upstream signal would be observed at the Receiver Site. In the models, the gRNA location was set on the top strand. **c** The design of validation PCR to identify the genomic sequence of the Donor Sites. Two specific primers (Nest-F1 and Nest-F2) were designed upstream of the gRNA of the Receiver Site. The Nest-F1 and Nest-F2 were sequentially used with the downstream Tn5 primer, and two amplicons were generated. The second amplicons were sent for Amplicon-seq. **d** The translocation cases identified by PEAC-seq + Amplicon-seq. **e** Translocation scores of all sites were plotted. The red arrow indicated the Receiver Site in Fig. 3d. A DNA translocation score was calculated as "translocation reads number"/("normal reads number" + " translocation reads number" + 10). Source data are provided as a Source data file.

with MMLV and employed the template information from pegRNA to label and enrich the editing sites. Utilizing the PEAC-seq, we successfully identified and validated off-targets both in HEK293T and in mouse embryos.

Recent studies have reported a variety of modifications to the native PE system to increase the editing efficiency[20–22]. We demonstrated that incorporating epegRNA is the most effective method to improve the insertion efficiency of PEAC-seq tags, which also rescued two missing off-targets from *EMX1* PEAC-seq (Fig. 5c, d). It is not surprising that the transient expression of MLH1dn did not improve the performance, as MLH1dn is a dominant negative MMR protein, which involves DNA heteroduplexes by selectively replacing nicked DNA strands[22]. However, the repair pathway activated by PEAC-seq is probably different, as we used the wild-type Cas9 to replace the Cas9 nickase in the native PE system.

Besides the off-targeting indels, DNA translocation happens when multiple DSBs were introduced and is more toxic to the genome stability[14,23]. Multiple DSBs might be introduced when a single gRNA was used but off-target editing happen, or when multiple gRNAs were used. For example, to engineer T cells to become allogeneic or autogeneic CAR-T, more than one gRNA needs to be used[12,13,24]. These further urged a sensitive translocation detection method to systematically profile DNA translocations. Recently, several papers reported that DNA translocations happened more frequently than we thought during Cas9 editing in vivo[25–27]. To our knowledge, besides ultra-deep whole-genome sequencing, none of the CRISPR off-target detection techniques are able to directly detect the DNA translocations without knowing the sequence of at least one DSB end. GUIDE-seq reported large-scale genomic alterations via AMP (anchored multiplex PCR)-based sequencing, in which a candidate translocation could be detected in the following validation step[5]. The directional insertion sequence in PEAC-seq allowed us to identify the aberrant ends joining from different DSB sites. We also noticed that the occurrence of DNA translocation is independent of the frequency of DSB at a particular site, which indicated that other factors, e.g., position or DSB context sequences might contribute to translocation[11]. Finally, due to the potential genotoxicity of the DNA rearrangements, both the translocation profiling methods and genotoxicity assessment need to be developed for CRISPR transitional applications.

PEAC-seq also conducted proof-of-concept studies to demonstrate its application in vivo. This method, together with DISCOVER-seq, both relying on agent signals that accompany the cleavage events. DISCOVER-seq used MRE11 ChIP-seq signals to represent the DSB events undergoing in the edited cells, while the nature of the ChIP-seq technique captured only the snapshot of MRE11 binding and might not exhibit the off-target sites over the course of editing. PEAC-seq, which relies on the enrichment of inserted PCR handle, might also overlook cleavages with incomplete insertions that could not be effectively enriched, although our random sequence screen demonstrated good efficiency of long insertion. Increasing the size of cell population might further increase the sensitivity of PEAC-seq, which has been demonstrated by the two verified PEAC-seq unique off-targets *in cellula*. Nevertheless, these methods, together with previous approaches,

provided versatile tools to enhance our understanding of the occurrence of off-target in different contexts, which are very informative alternatives to the costly WGS.

Finally, it is intrinsically interesting that not all potential off-target sequences are eventually edited as off-targets. To look into this question, we analyzed the genomic co-localizations between the PEAC-seq off-targets and epigenetic signals collected from public data[28]. We plotted the density of ATAC-seq peaks and ChIP-seq peaks of multiple histone modifications and proteins surrounding (±5 kb) the PEAC-seq off-targets. Briefly, the results indicated that off-targets tended to occur in open chromatin regions (ATAC-seq) and to be associated with histone modifications in active gene regulation (H3K4me3, H3K9ac, and H3K27ac) and gene transcription (POLR2A, EP300, H2AFZ) (Fig. 6a). PEAC-seq translocation does not associate with the above epigenetic marks as well as cancer-related fusion genes, but show co-occurrence with the double-strand breaks (DSBs) in HEK293T cells. Compared to control regions, which were equally sized regions re-sampled randomly across the genome, we observed enrichment of DSBs surrounding ±5 kb of the PEAC-seq translocation sites (Fig. 6b), indicating that CRISPR editing-induced translocation tends to occur at DSB-enriched regions.

The limitation of this study, however, is that the insertion efficiency of the PEAC-seq tag might vary across different pegRNAs and at different off-targets. For each pegRNA, the RNA secondary structure of the insertion tag and sequence uniqueness to the host genome could vary. But if the aforementioned guidelines were taken into account, this sequence is interchangeable, and we have supplied a few additional tested sequences (Table S11 and S12). Regarding the PBS (primer binding site) length, we inherited a 13-nt design according to the native PE system[18], although both the 13-nt and 17-nt worked equally well in our hands. And the PBS sequences, which were derived from the on-target sites, can be different at off-target sites. Mismatches between the PBS and the spacer sequences at off-target sites might affect primer extension in the reverse transcription and result in low insertion efficiencies of the PEAC-seq tag. Actually, the two missing off-targets in the *VEGFA TS1* site include PBS mismatches at positions 14 and 17 (5' to 3') at the off-target sites (Fig. 1c and Table S10), which are proximal to the starting point of primer extension of reverse transcription (Supplementary Fig. 12a). GUIDE-seq-unique off-targets in the *VEGFA TS2* and *VEGFA TS3*, not verified by Amplicon-seq though, also contained relatively more PBS mismatches compared to the shared and the PEAC-seq-unique off-targets (Supplementary Fig. 12b, c). However, many off-targets with PBS mismatches were successfully identified by PEAC-seq, indicating the complication of the effects of PBS mismatches on reverse transcription. Nevertheless, we propose to include a few random nucleotides in the PBS regions of pegRNA (mut-pegRNA) (e.g., proximal to the primer extension site) to improve the extension efficiency at off-targets with PBS mismatches (Supplementary Fig. 13). According to this study's PEAC-seq and ePEAC-seq data, pegRNA designed from the on-target sequence could enable PEAC-seq tag insertion in most off-target sites, and the

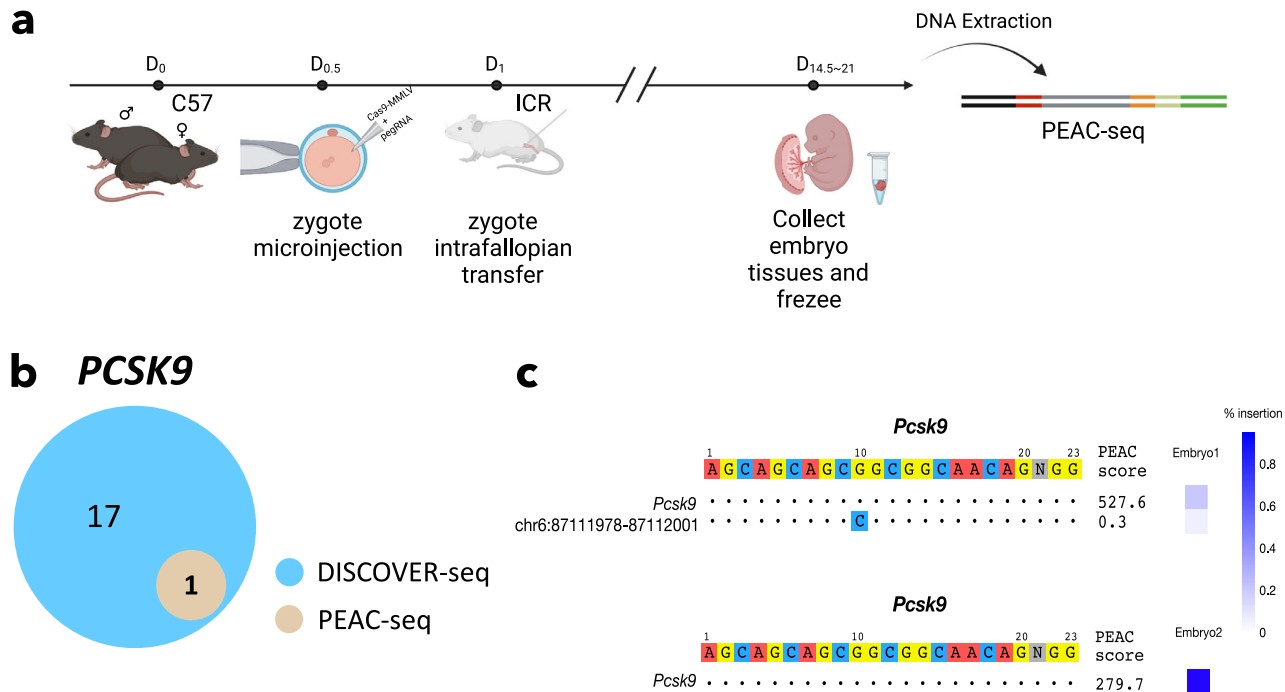

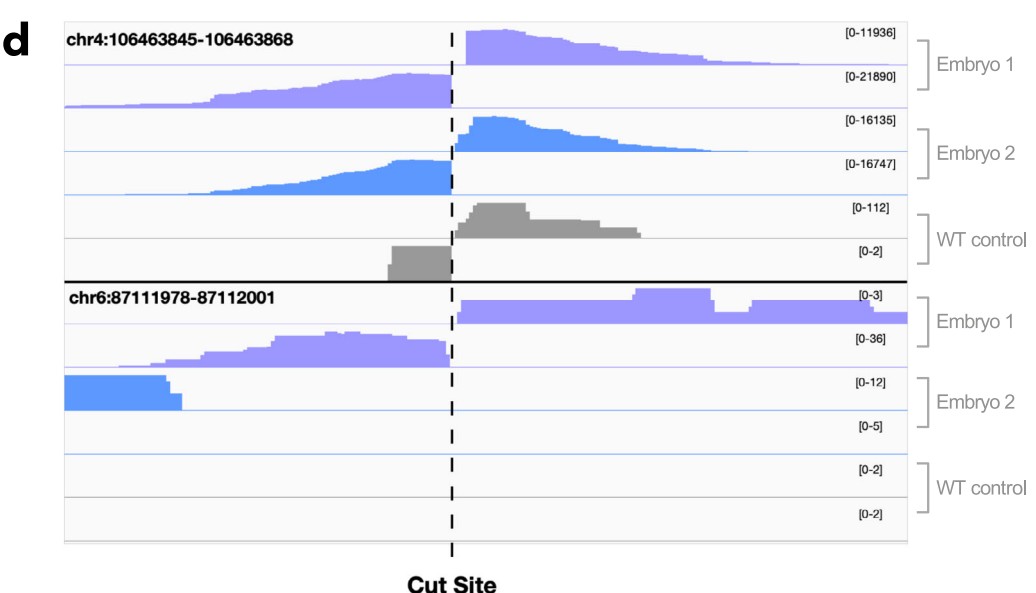

**Fig. 4 | PEAC-seq identified pcsk9 off-targets from an edited mouse embryo.**
**a** Schematic representation of the in vivo PEAC-seq experiment. **b** The Venn diagram shows the overlap between the PEAC-seq on-target and off-targets of *PCSK9* and the top18 editing sites (including the on-target) identified by DISCOVER-seq. **c** The sequence visualization of the *PCSK9* on-target and off-targets. One off-target was identified from one of the two embryos. The site was also reported by DISCOVER-seq and validated by Amplicon-seq. The color scale represented the indel frequency reported by CRISPResso. **d** The signal track of the on-target and off-target sites identified from PEAC-seq in two different embryos and wild-type control. The signal of the WT control at chr4:106463845 was 1000-fold lower than the sample and was considered as background. Source data are provided as a Source data file.

incorporation of mut-pegRNA might improve the insertion efficiency of PEAC-seq tags in some off-target sites with critical PBS mismatches. Besides, reverse transcriptase evolving for error-correcting activity (e.g., error-correcting reverse transcriptase[29]) may further improve the primer extension efficiencies. If proper enzyme could be evolved and characterized, the 3′ to 5′ exonuclease activity could correct mismatches between PBS and off-targets.

In summary, we adopted the Prime Editor system to report CRISPR off-targets *in cellula* and in vivo, and Cas9-dependent DNA rearrangement. Using pegRNA to provide a sequence-optimized template, PEAC-seq further diversified the CRISPR off-target identification toolbox and provided a reliable solution to directly identify off-targets for in vivo editing and recognize DNA rearrangements, which both would strengthen our ability to assess the genotoxicity in CRISPR therapies.

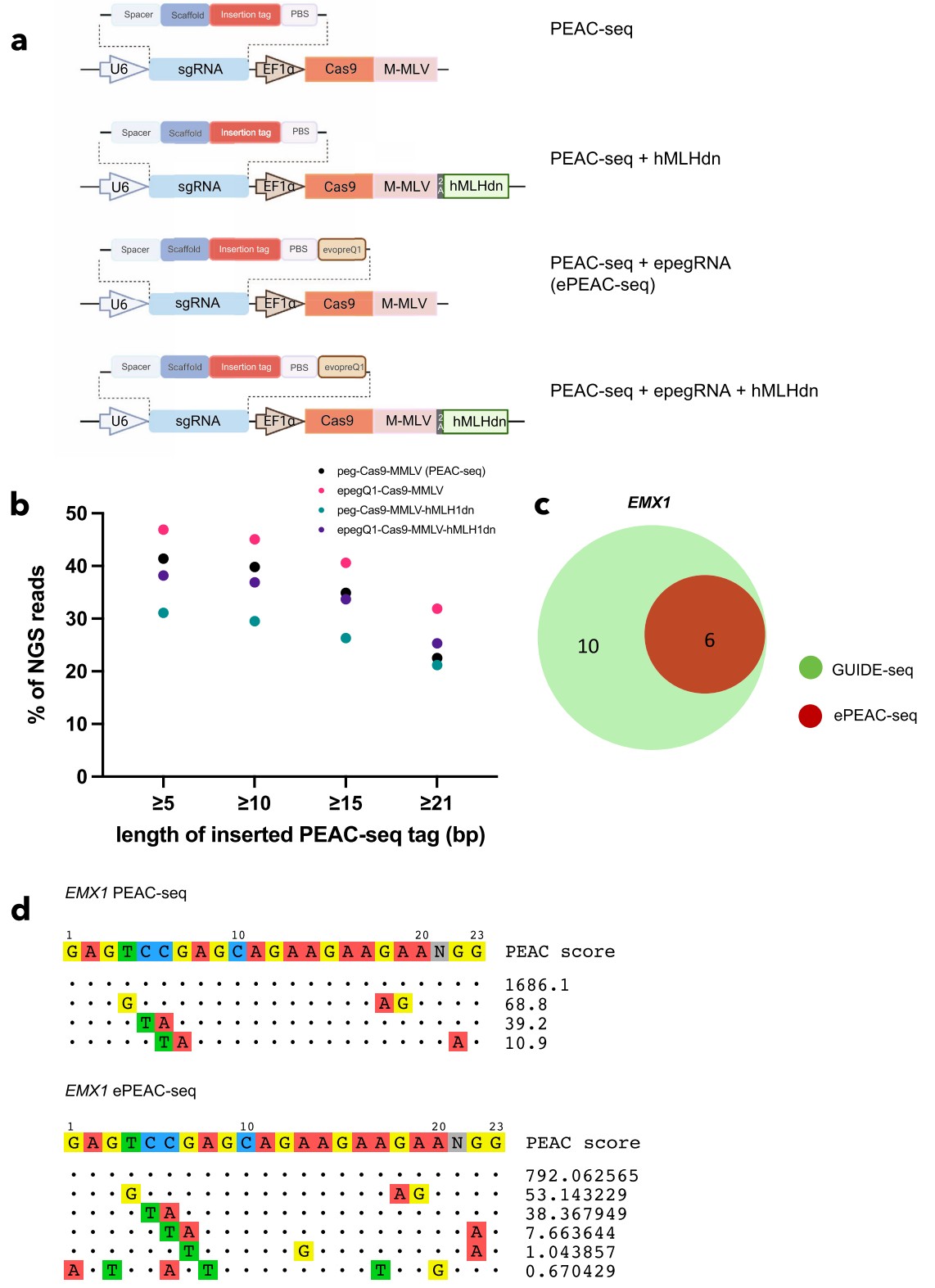

**Fig. 5 | ePEAC-seq is an enhanced version of PEAC-seq with higher sensitivity to identify off-targets. a** Schematic representation of the three modified versions of PEAC-seq. **b** The insertion frequencies of PEAC-seq tag in PEAC-seq and its three

modifications. **c** The Venn diagram of *EMX1* off-targets identified by PEAC-seq and GUIDE-seq. **d** ePEAC-seq identified two more verified off-targets that were missed by PEAC-seq. Source data are provided as a Source data file.

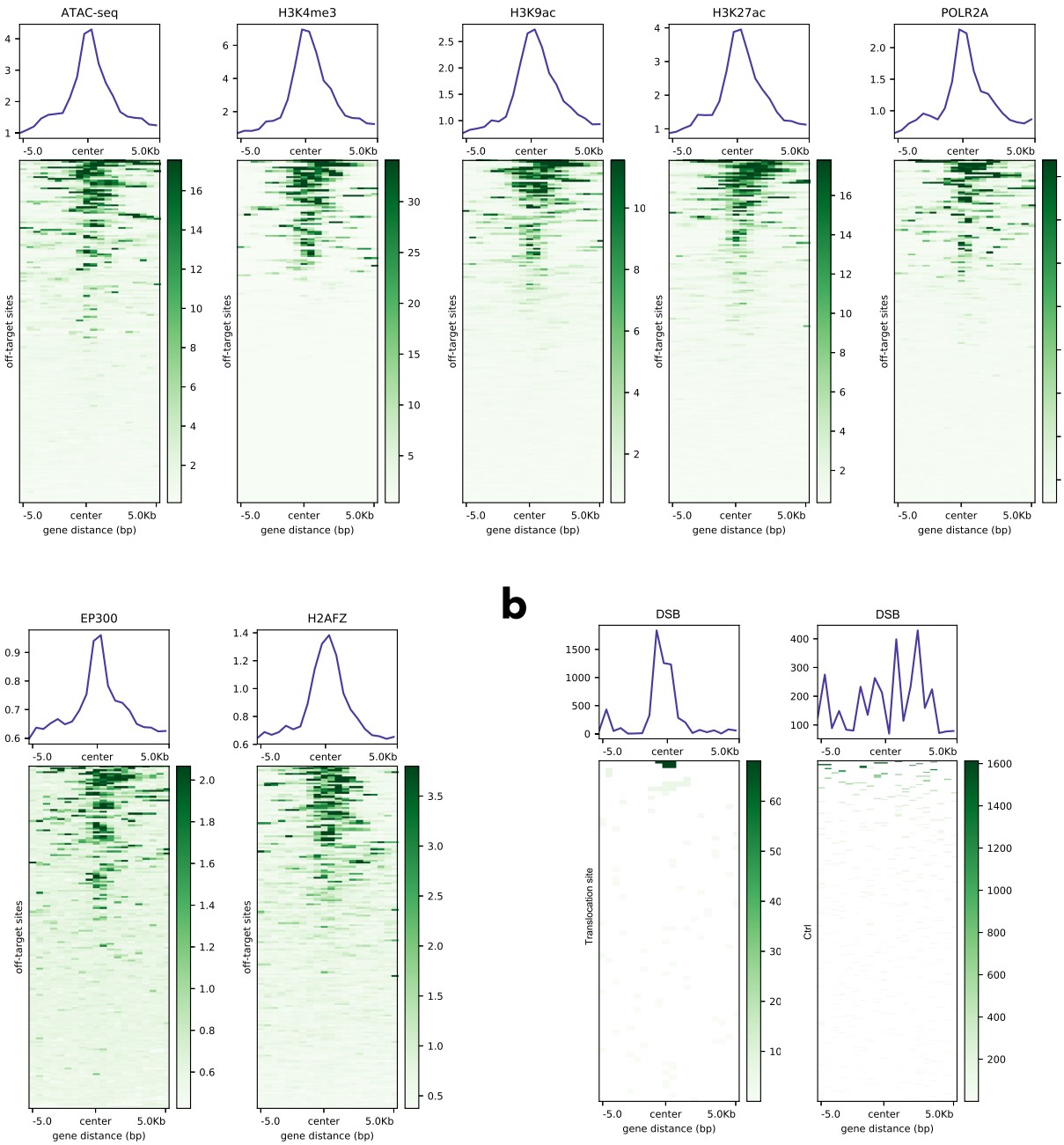

**Fig. 6 | The genomic context of PEAC-seq off-target and translocations.**
**a** Signals of the ATAC-seq peaks and ChIP-seq peaks of multiple histone modifications and proteins surrounding the PEAC-seq off-targets. **b** Signals of the DSB surrounding the PEAC-seq translocation sites (left panel) and random controls (right panel).

## Methods

### Ethical statement

The animal experiments of this study comply with animal protocols approved by the Laboratory Animal Resource Center (LARC) at Westlake University.

### PEAC-seq in HEK293T cell

We adopted the Prime Editor system by replacing the Cas9 nickase with wildtype Cas9 and modified the RT template of pegRNA, and assembled them into a single vector as the PEAC-seq backbone. The spacer sequences targeting *VEGFA*, *EMX1*, *RFN2*, and *FANCF* were cloned into the PEAC-seq backbone individually. To conduct PEAC-seq in living cells, HEK293T cells were seeded in a 12-well plate and grow till ~80% confluency. Each well was transfected with 3 µg plasmids by Lipofectamine 3000. The post-transfection cells were collected after 48 h. The cell sorter (SONY MA900) was used to sort about 100,000 GFP-positive cells (Supplementary Fig. 14). About 500 ng extracted gDNA was digested with NotI then cleaned up with 0.5× AMPure XP beads to remove the carryover plasmids. The gDNA fragments were retained on the AMPure XP beads, and on-beads Tn5 digestion was performed at 55 °C for 1 h, and adapters were inserted at the ends of the fragments. The Tn5 was expressed and embedded with the adapters in-house. At the end of the Tn5 digestion, 6 µL 0.2% SDS was added to terminate the reaction. The products were purified and size-selected

by 1.5× AMPure XP beads and eluted in 50 μL H$_2$O. The 21 bp insertion sequence was used to enrich the editing sites (both on-target and off-target) in the NGS library preparation. In the first round of the nested PCR, two separate reactions were performed. Each reaction used a 20 μL template in a total of 50 μL volume at ~30 cycles. One used the PEAC-seq insertion sequence as the forward primer binding site and the downstream Tn5 adapter as the reverse primer binding site. Another used the upstream Tn5 adapter as the forward primer binding site and the PEAC-seq insertion sequence as the reverse primer binding site. In all, 2.5 μL first round product was used as the template in the second round amplification in a total of 50 μL volume at 17 cycles, and Illumina adapters were added. The amplicons were purified by AMPure XP beads using 0.6× + 0.25× double size selection. The library was sequenced on the Illumina Novaseq platform as paired-end 150 bp.

The oligo and vectors are summarized in Supplementary Data 9.

## Translocation characterization

To identify the translocated sequences, we designed two nested PCR primers upstream of the gRNA. The site-specific nested PCR primers served as forward primers, and downstream Tn5 primer served as reverse primer. The nested primers were sequentially used to amplify the adjacent sequences of translocated DSBs. About 300 ng PEAC-seq gDNA was fragmentized by Tn5, purified with 1.5× AMPure XP beads and eluted with 23 μL H$_2$O. About 20 μL purified DNA was used as template for the first round PCR for 20 cycles. And 2.5 μL products from the first PCR was used as template for another 20 cycles in the second round of the nested PCR. Another 20-cycle PCR was conducted to add the sequencing adapters. The amplicons were purified by 0.6× and then 0.25× double-size beads selection. The library was sequenced on the Illumina Novaseq platform as paired-end 150 bp.

In the DNA translocation analysis, we summarized the reads number and reads orientation from the forward and backward PCR libraries around the on-target and candidate off-target sites. A DNA translocation score was calculated as "translocation reads number"/ ("normal reads number" + "translocation reads number").

The oligo and vectors are summarized in Table S9.

## In vivo off-target detection by PEAC-seq

Both the pegRNA and the mRNA of Cas9-MMLV were prepared by in vitro transcription. The DNA template of pegRNA was amplified from the plasmids "pcsk9-sgRNA" and "mPnpla-sgRNA" by primers T7F and T7R. The PCR products were gel purified using MinElute Gel Extraction Kit (QIAGEN #28606), which was used as the template for in vitro transcription by HiScribe T7 Quick High Yield RNA Synthesis Kit (NEB #E2050S). The pCMV-Cas9-PE2 plasmid was linearized by MssI (Thermo #FD1344). According to the manufacturer's instructions, 1 μg linearized product was used as a template to generate Cas9-PE mRNA from in vitro transcription by HiScribe T7 ARCA mRNA Kit (NEB #E2060S).

C57BL/6 and ICR mice were purchased and housed in the Laboratory Animal Resource Center (LARC) at Westlake University. The LARC is a certified pathogen-free and environmental-control facility (21 ± 2 °C, 55 ± 15% humidity, and 12:12-h light:dark cycle). The C57BL/6 mice were used for embryo collection, and ICR females were used as recipients. All animal experiments were conducted under the protocol approved by the animal care and ethical committee of Westlake University.

Six-week-old C57BL/6 female mice were superovulated by injecting 5 IU of PMSG (Pregnant Mare Serum Gonadotropin; ProSpec #HOR-272), then followed by 5 IU of hCG (human chorionic gonadotropin; ProSpec # HOR-250) after 48 h. The C57BL/6 females were then mated to 8-week-old C57BL/6 males. After 16 h, fertilized embryos were collected and placed in EmbryoMax M2 Medium with Hyaluronidase (Millipore #MR-051-F). After the cumulus cells fell off, embryos were transferred into a dish containing 2 mL of fresh M2 medium (Millipore #MR-015-D). Embryos were then flushed several times to rinse off the hyaluronidase and cumulus cells. Afterward, embryos were transferred into a dish with prewarmed KSOM medium (Millipore #MR-106-D) covered by mineral oil followed by three additional washes.

The mixture of Cas9-PE2 mRNA (100 ng/μL) and pegRNA (50 ng/μL) was injected into the cytoplasm of the zygote in M2 medium. The injection was conducted using a microinjector (NARISHIGE #IM-400B) with constant flow settings. The injected embryos were cultured in KSOM medium with amino acids in a cell culture incubator at 37 °C and with 5% CO$_2$, then were transplanted into oviducts of pseudopregnant ICR females at 0.5 dpc. Pups were sacrificed at E19.5–E21, and organs were collected, dissected, and snap-frozen in liquid nitrogen. Samples were stored at −80 °C until further analysis.

The gDNA from organs was extracted using TIANamp Genomic DNA Kit (TIANGEN #DP304-03) according to the manufacturer's instructions. Nested PCR was applied to amplify the targeting regions and attach the Illumina adapters to amplicons. The in vivo PEAC-seq library was constructed as the cell line data in the previous section by Tn5 fragmentation.

The oligo and vectors are summarized in Supplementary Data 10.

## Data analysis

The PEAC-seq data were analyzed using a modified pipeline from GUIDE-seq[5]. Firstly, we trimmed adapters using cutadapt[30], and reads without appropriate adapter were removed. Then the reads were mapped to the human or mouse genome (hg38, mm10) using bwa. Reads mapped to the same location and shared the same UMI were considered PCR duplicates and merged in the following analysis. In order to fit in the target identification pipeline from GUIDE-seq, the reads name from bam files was modified, and the bam files from the forward and backward PCR were labeled and merged. Modifications were made to the pipeline to remove reads originating from random priming. In summary, we normalized the reads number from the GUIDE-seq output file to reads per million and calculated the number of reads with correct primer extension. The candidate sites meet the following criteria: (1) no signal in the wild-type control sample; (2) the number of reads with correct primer extension sequence ≥1 at least in one direction, and the geometric mean of the primer extension reads >0; (3) correct reads strand information on both the upstream and downstream of the putative gRNA cutting site. The Amplicon-seq data was analyzed using CRISPResso 2.13 (--max_paired_end_reads_overlap 140 --min_paired_end_reads_overlap 10 --exclude_bp_from_left 0 --exclude_bp_from_right 0 --plot_window_size 40 --min_frequency_alleles_around_cut_to_plot 0.1)[31].

## Visualization of the genomic co-localizations between the PEAC-seq off-targets and epigenetic signals

The bigWig files of ATAC-seq and ChIP-seq datasets from HEK293T cells (H3K4me3, H3K9ac, H3K27ac,POLR2A, EP300, H2AFZ) were downloaded from epimap (https://epigenome.wustl.edu/epimap/data/imputed/)[28]. Deeptools "computeMatrix" (command: --referencePoint center --afterRegionStartLength 5000 --beforeRegionStartLength 5000 -p 15 --binSize 500) and "plotHeatmap" function[32] were used to visualize the genomic co-localizations between all in vitro PEAC-seq off-target sites and epigenetic signals. DSBs hotspots were identified from the dsODN only control (no Cas9/gRNA) from the GUIDE-seq performed in the 293T cells. Control genomic regions, which were equally sized regions randomly across the genome, were generated with the in-house perl script. Deeptools "computeMatrix" and "plotHeatmap" function were used to plot the heatmap of the genomic co-localizations between the translocation sites and DSBs.

## Statistics and reproducibility

No statistical method was used to predetermine sample size. No data were excluded from the analyses. The experiments were not

randomized. The Investigators were not blinded to allocation during experiments and outcome assessment.

### Reporting summary

Further information on research design is available in the Nature Portfolio Reporting Summary linked to this article.

## Data availability

The authors declare that all data supporting the findings of this study are available within the paper and its Supplementary Information files. High-throughput sequencing data supporting this study has been deposited into the Gene Expression Omnibus (GEO) database (NCBI). It is accessible under GEO Series accession numbers GSE179523 and GSE179436. Source data are provided with this paper.

## Code availability

The supported PEAC-seq analysis code has been uploaded on the GitHub website: https://github.com/LijiaMALab/PEACSeq.

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

## Acknowledgements

We thank the Flow Cytometry Facility and Genomics Facility of the Westlake Biomedical Research Core Facilities and Laboratory of Animal Resource Center of Westlake University for their assistances in conducting this work. We thank Dr Dangsheng Li for valuable discussions and suggestions. Cartoons in Figs. 1a, b, 4a, 5a, and S1 were created with BioRender.com. The plots were created by graphpad prism8 and excel. This research was supported by Center for Genome Editing, Westlake Laboratory of Life Sciences and Biomedicine [program No. 21200000A992210/003 to L.M.], Basic Research Foundation of Zhejiang Province for Distinguished Young Scholar [LR21C060001 to L.M.], National Key R&D Program of China [2021YFC2700804], Westlake University Industries of the Future Research Funding [210210006022208 to L.M.], National Science Foundation of China [31950575 to L.M.], and Startup Funding to L.M. from the Westlake Education Foundation.

## Author contributions

L.M., Z.L., and Z.Y. conceived the project and designed the experiments. Z.Y. and J.L. performed experiments with the help of P.W., B.L., H.Z., and Y. Liu. Z.L. and Y.W. analyzed the data with the help of Y. Li and Y.Z. L.M., Z.Y., Z.L., Y.W., and J.L. wrote this paper.

## Competing interests

L.M., Z.L., and H.Z. are co-founders of AIdit, a biotech startup focusing on AI-assisted CRISPR therapy. The remaining authors declare no competing interests.
