## [Peer Review File · Nature Communications]

Reviewers' Comments:

Reviewer #1:

Remarks to the Author:

In this manuscript, the Authors describe a novel method named PEAC-seq to detect undesired off-target mutations and translocations associated with CRISPR genome editing. The Authors devised a clever combination of the Prime Editor system and the Tn5 transposase to selectively amplify and sequence the genomic sites where a specific DNA cassette is inserted by Cas9 fused to the Moloney Murine Leukemia Virus (MMLV) Reverse Transcriptase.

The Authors benchmarked PEAC-seq against widely used methods for CRISPR off-target detection—such as GUIDE-seq, DISCOVER-seq, and targeted amplicon sequencing—and assessed the sensitivity and specificity of their new approach both in cultured cells and in mouse embryos.

In my opinion, the major limitation of this manuscript is that, like many other published methods on CRISPR off-target detection, the Authors do not go beyond providing a mere list of off-targets. For example, the Authors report that the frequency of translocations is substantially higher at certain off-target sites compared to the on-target as well as other off-target sites. However, the Authors do not provide any in-depth characterization of these sites, which is critical to understand the basis of off-target formation.

Below I enclose a series of remarks and suggestions to help the Authors revise their manuscript.

MAJOR REMARKS

- 1) In its present form, the manuscript is very difficult to read due to the presence of multiple grammatical errors and heavy syntax. I strongly recommend that the Authors consider having a native English speaker edit their manuscript. (To be clear, this Reviewer is not a native English speaker and is by no means attempting to discriminate the Authors based on their knowledge of English).
- 2) The amplification strategy depicted in Fig. 1a and 1b is not well explained: what is the gray sequence surrounding the PEAC-seq insertion tag? Is this sequence known and used to design the F1-3 primers? What are the overhangs in the F1-3 and R1-2 primers? Also, in the last step of the scheme shown in Fig. 1a, are two separate PCR reactions performed? Or are the two F-R primer pairs shown used in a single multiplex PCR reaction? The Authors should better explain the details of their approach and also describe in the Introduction how the Prime Editor system works.
- 3) Similarly, the schemes shown in Fig. 3a and 3b are extremely difficult to understand, in part due to the colors used (the strands colored in red and orange are almost indistinguishable). What is the difference between receiver and donor site? Are these two sites targeted with two distinct pegRNAs? Or do the Authors refer to 'Receiver' and 'Donor' as the two DSB moieties that are fused together in a translocation? The Authors should explain this part in a much clearer manner and improve the schemes in Fig. 3a, b accordingly.
- 4) Regarding the detected translocations, the Authors should perform whole-genome sequencing on the edited cells to check if the most frequent translocations can be detected (and hence validated) and to assess whether other translocations that are not detected by PEAC-seq form upon editing. This is essential to gauge the sensitivity of PEAC-seq in detecting off-target translocation events. Did the Authors only detect translocations between different chromosomes or also inversions within the same chromosome? Can PEAC-seq detect inversions?
- 5) A major limitation of this study, like all previous studies describing new methods for CRISPR off-target detection, is that the Authors only provide a list of off-targets and compare the number of off-targets detected by different methods. Instead, it would be much more relevant to focus on where, along the genome, off-target events occur: is there a specific sequence or epigenome context that favours the formation of off-target alterations? The Authors should intersect their translocation data with a variety of available epigenome tracks (e.g., ATAC-seq or histone mark

ChIP-seq data from the same or related cell lines) trying to understand whether there exist specific features that prime these regions to undergo unwanted editing. Even more importantly, the Authors should investigate if the observed translocations affect cancer-related genes and compare off-target translocation breakpoints with those of cancer translocations found in human cancers (e.g., using the TCGA gene fusion database).

ADDITIONAL REMARKS

- 1) Throughout the manuscript and in figures, it is never specified in which cell line(s) the experiments were conducted (HEK293T?). Please provide this important information.
- 2) The Authors often use the expression in *cellulo* to describe experiments conducted in cultured cells. Although this expression is often encountered in scientific articles, it is grammatically incorrect because in Latin the word 'cell' (*cellula*) is female. Therefore, the correct expression is 'in *cellula*'.
- 3) Figure panel labels should be in small letters (e.g., Fig. 1a, b, c) as per Nature Communications style.
- 4) Fig. 1e: I assume that the schemes on the right indicate the strand and orientation of the PEAC-seq cassette (red). However, this is not explained and is different than what is shown in Fig. 1a, b where both strands of the cassette are colored in red. The Authors should clarify this, both in the figures and in the main text.
- 5) The Authors should consider providing supplementary BED files allowing the visualization of read pileups at on- and off-target sites, instead of showing all of them in Supplementary Figures that extend beyond one page.

Reviewer #2:

Remarks to the Author:

General comments;

CRISPR-based genome editing tools have widely been used for precise gene knockout and correction. But, nucleases-based potential limitations such as off-target editing, large deletion or chromosomal rearrangements have been continuously raised. To identify genome-wide off-target sites is a steady important issue in the genome editing field. In the present study, Yu et al developed PEAC-seq by adopting the prime editing technology (with not Cas9 nickase but wtCas9) to detect Cas9-mediated off-target and DNA translocation sites. Basically the PEAC-seq might be another in vivo approach for off-target searching, but I would like to raise a critical issue on the efficacy of this tool as follows.

1. I am skeptical on that the wtCas9 based prime editing may not have editing activity as much as the wtCas9 nucleases. For example, when ten arbitrary sites are selected, wtCas9 typically shows editing (or indel) frequencies about 50%, while wtCas9 based prime editing might show much less efficiencies (less than ~ 10%), which potentially indicates that the PEAC-seq could underestimate the off-target sites. I would like to emphasize that underestimation of off-target sites would be much worse than overestimation of them, in terms of a safety issue.

1-1. It is necessary to prove that the PEAC-seq does not generally underestimate the off-target sites compared to previous tools (i.e., other in vivo based tools such as GUIDE-seq, Discover-seq, Site-seq) with at least 3 different gRNAs in the MAIN Figure. One gRNA shown in Figure 1C is not enough. I concern that the PEAC-seq might miss several bona-fide off-target sites as shown in the Supplementary Figure 6.

2. The incorporate tag sequences by the PEAC-seq would be very critical but the information of it is not mentioned in the main text. What is the length of it? It is well known that the editing efficiency is negatively correlated to the length of the insertion sequences. Hence, when the

incorporate tag is about 20-bp length, the precise incorporation efficiency might be very low even at the on-target site.

If then, the precise incorporation efficiency at off-target sites would be extremely low, indicating that the PEAC-seq is not a sufficient tool for searching genome-wide off-targets.

3. Prime editing efficacies vary according to the RTT and PBS lengths. The strategy to design a optimized pegRNA for the PEAC-seq should be suggested.

(minor)

- In the title, 'adapt' is correct? 'adopt' might be possible.
- In page 4, 'Supplementary FigX' should be corrected.
- It is not easy to understand the meaning of the Figure 2C. Do inserted positions vary within gRNA?
- In the discussion section, the authors argued that the PEAC-seq is the only tool for detecting translocation sites, but HTGTS is a dedicated tool for detecting CRISPR-mediated translocation which was already published in 2015 [PMID: 25503383].

Response to Reviewer #1

In this manuscript, the Authors describe a novel method named PEAC-seq to detect undesired off-target mutations and translocations associated with CRISPR genome editing. The Authors devised a clever combination of the Prime Editor system and the Tn5 transposase to selectively amplify and sequence the genomic sites where a specific DNA cassette is inserted by Cas9 fused to the Moloney Murine Leukemia Virus (MMLV) Reverse Transcriptase.

The Authors benchmarked PEAC-seq against widely used methods for CRISPR off-target detection—such as GUIDE-seq, DISCOVER-seq, and targeted amplicon sequencing—and assessed the sensitivity and specificity of their new approach both in cultured cells and in mouse embryos.

In my opinion, the major limitation of this manuscript is that, like many other published methods on CRISPR off-target detection, the Authors do not go beyond providing a mere list of off-targets. For example, the Authors report that the frequency of translocations is substantially higher at certain off-target sites compared to the on-target as well as other off-target sites. However, the Authors do not provide any in-depth characterization of these sites, which is critical to understand the basis of off-target formation.

Below I enclose a series of remarks and suggestions to help the Authors revise their manuscript.

MAJOR REMARKS

1) In its present form, the manuscript is very difficult to read due to the presence of multiple grammatical errors and heavy syntax. I strongly recommend that the Authors consider having a native English speaker edit their manuscript. (To be clear, this Reviewer is not a native English speaker and is by no means attempting to discriminate the Authors based on their knowledge of English).

Response: We apologize for the grammatical errors and syntax in the originally submitted manuscript. As the Reviewer suggested, we have asked a native English speaker to edit the revised manuscript.

2) The amplification strategy depicted in Fig. 1a and 1b is not well explained: what is the gray sequence surrounding the PEAC-seq insertion tag? Is this sequence known and used to design the F1-3 primers? What are the overhangs in the F1-3 and R1-2 primers? Also, in the last step of the scheme shown in Fig. 1a, are two separate PCR reactions performed? Or are the two F-R primer pairs shown used in a single multiplex PCR reaction? The Authors should better explain the details of their approach and also describe in the Introduction how the Prime Editor system works.

Response: Thanks for your comments on the visualization of the PEAC-seq

scheme. As suggested, we have made modifications to the previous plot and rewritten the figure caption to improve the clarity:

- 1) We added labels to the revised Fig. 1a to indicate the regions of the insertion tag sequence and the surrounding sequences (**Response Document Figure #1a**).
- 2) In the revised Fig. 1a, Fig. 1b, and Fig. 2c, we used consistent colors to indicate the insertion tag sequences and the surrounding sequences (**Response Document Figure #1a, #1b, and #1c**).
- 3) In caption of revised Fig. 1b, we described where the F1-3 primers and the R1-2 primers were primed and how did the last PCR step was conducted (**Response Document Figure #1b**).

Response Document Figure #1 (presented as Fig. 1a, 1b, and Fig. 2c in the revised manuscript)

The revised figure caption was quoted as follow:

“Fig. 1 Development of the PEAC-seq technique.

a. Schematic representation of the PEAC-seq experimental procedure. The gDNA were extracted and undergone Tn5 tagmentation. The Tn5 was embedded with UMI-adaptors to eliminate PCR duplications. After tagmentation, fragments were amplified by pairs of primers (one priming at the PEAC-seq insertion, another priming with the Tn5 adaptor).

b. Schematic representation of the two forward primers and two reverse primers designed for tag enrichment and library preparation of PEAC-seq. Each forward primer was paired with a downstream Tn5 primer to generate amplicons including the PEAC-seq tag sequence and its downstream genomic sequences. Each reverse primer was paired with an upstream Tn5 primer to generate amplicons including

the PEAC-seq tag sequence and its upstream genomic sequences. In total, five Amplicon-seq data from the three forward primers and two reverse primers were generated, and six candidate lists of putative off-targets were inferred from the five Amplicon-seq data using a modified GUIDE-seq analysis pipeline (**Methods**).

Fig. 2 Analysis on the PEAC-seq off-target sites.

c. Screenshots of PEAC-seq signal tracks from the IGV Genome Browser. One on-target site, one shared off-target site, and one PEAC-seq unique off-target site were presented. For each site, signals from both the PEAC-seq and the wild-type (WT, no Cas9-MMLV treatment) samples were included. For each sample, the first track represented signals from the amplicons of a forward primer and a downstream Tn5 primer; the second track represented signals from the amplicons of a reverse primer and an upstream Tn5 primer. The model on the right side showed the direction of spacer and PAM of each case.”

- 4) We have included introduction content about the Prime Editor system in the Introduction of the revised manuscript.

“Here, we introduced a new off-target identification method, PEAC-seq (Prime Editor Assisted off-target Characterization), in which we designed a Cas9-MMLV fusion protein to take advantage of the sequence insertion ability from the Prime Editor (PE)¹. The native PE system (Cas9n-MMLV) utilizes a pegRNA (Prime Editor gRNA) containing extra sequences at the 3' of gRNA, which serve as a priming site and allow reverse transcription (RT) from the exposed 3'-hydroxyl group of the non-targeting strand to incorporated additional DNA sequences into the editing sites.”

3) Similarly, the schemes shown in Fig. 3a and 3b are extremely difficult to understand, in part due to the colors used (the strands colored in red and orange are almost indistinguishable). What is the difference between receiver and donor site? Are these two sites targeted with two distinct pegRNAs? Or do the Authors refer to 'Receiver' and 'Donor' as the two DSB moieties that are fused together in a translocation? The Authors should explain this part in a much clearer manner and improve the schemes in Fig. 3a, b accordingly.

Response: Thanks for your comments on the visualization of the PEAC-seq scheme. To improve the clarity, we have made the following modifications to the previous plot.

- 1) We have changed the colors of the 'Receiver' and 'Donor' sequences for better visualization (**Response Document Figure #2**).

2) We have explicated our intended meaning for ‘Receiver’ and ‘Donor’ sequences in both the figure caption and the main text. The “Receiver” and “Donor” are the two DSB moieties that are fused together in a translocation. The “Receiver” is the moiety that located at the upstream of the PEAC-seq insertion tag, and the “Donor” is the moiety that was brought to fuse with the “Receiver” from another genomic location.

Response Document Figure #2 (presented as Fig. 3b in the revised manuscript) The diagrams showing three DSB ends induced by PEAC-seq. (1) upstream of DSB; (2) upstream of DSB that incorporated with PEAC-seq insertion tag; (3) downstream of DSB. Both Receiver (top left) and Donor (top right) sites could generate the three DSB ends. The upstream of DSB of a Receiver site could be joined with its own downstream of DSB by DNA repair (bottom left panel, models (i) and (ii)); The upstream of DSB of a Receiver could also be fused with each of the three DSBs from a Donor site (bottom left panel, models (iii), (iv), and (v)). When PEAC-seq insertion tag presented, signal tracks from PEAC-seq experiments were shown in the bottom right (only amplicon signal between F primer and downstream Tn5 primer were shown).

The revised figure caption was quoted as follow:

“Fig. 3 PEAC-seq identified DNA translocations relevant to CRISPR genome editing

b. Proposed models of the generation of unexpected upstream signals. Both the Receiver site and the Donor site could generate DSBs and proximal to each other within the nucleus. Model (i) and Model (ii) joined DSB ends from the same Receiver site. Model (iii), Model (iv) and Model (v) joined one donor DSB and one Receiver DSB. If the donor DSB carried the PEAC-seq insertion, the unexpected upstream signal would be observed at the Receiver Site. In the models, the gRNA location was set on the top strand.”

The revised text was quoted as follow:

“To enrich PEAC-seq tag, the forward primer (F1) and downstream Tn5 primer would amplify regions downstream, but not upstream, of the PEAC-seq tag (Fig. 3a). Surprisingly, in some cases, we saw unexpected signals located at the upstream genomic region of the F1-Tn5 amplicons (Fig. 3a, Supplementary Fig. 9). With further analysis on these sites, we speculated that the signals might come from the joining of DSB ends from another genome breaking sites. As shown in the proposed models (Fig. 3b), PEAC-seq generates DSBs with three different ends, including one upstream end appended with a complete or partial PEAC-seq tag, one upstream end without PEAC-seq tag, and one downstream end. If multiple DSBs simultaneously occurred in nucleus and physically proximal to each other, DSB ends from different breaking points might join together and cause DNA rearrangements. In our hypothesized scenario, the upstream end with the PEAC-seq tag from a distal Donor Site may join to the upstream end of a Receiver Site, but the direction of the PEAC-seq tag is reverse relative to the Receiver Site (Fig. 3b, model (v)). This joining generates signals upstream to the PEAC-seq tag of the Receiver Site, which won't be amplified by the F1 and Tn5 primers (Fig. 3a).”

4) Regarding the detected translocations, the Authors should perform whole-genome sequencing on the edited cells to check if the most frequent translocations can be detected (and hence validated) and to assess whether other translocations that are not detected by PEAC-seq form upon editing. This is essential to gauge the sensitivity of PEAC-seq in detecting off-target translocation events. Did the Authors only detect translocations between different chromosomes or also inversions within the same chromosome? Can PEAC-seq detect inversions?

Response: Thanks for focusing our attention here. By following the Reviewer's suggestions, we conducted 20x whole-genome sequencing (WGS) on the cells that were treated with the PEAC-seq cassette and were used to identify off-targets and translocations of the gRNA targeting VEGFA TS3 in the previous manuscript. A control WGS experiment was performed on wild-type HEK293T cells to distinguish

genome translocations that were not introduced by genome editing. The 20x sequencing successfully identified a translocation event between two off-targets of VEGFA TS3; both of these off-targets were identified by PEAC-seq.

We expect that more translocation events will be identified with deeper WGS². However, although WGS is an unbiased method to examine aberrant DNA sequences from a cell population with low genome heterogeneity (e.g., for a cell population grown from one clone), it is not sensitive to profile rare aberrant events that occur at low frequencies in a subpopulation of cells (e.g., translocation).

Thus, we validated PEAC-seq translocations using UDiTas, an enrichment-based method to detect indels and genome rearrangements after genome editing³ (**Response Document Figure #3**). Among the 43 PEAC-seq off-targets of VEGFA TS3, we identified 17 sites that were associated with genome translocation (presented as **Fig. 3e** in the revised manuscript). We performed UDiTas on two of the 17 sites with the highest “translocation score”, which was calculated as “num. of translocated reads / (num. of translocated reads + num. of normal reads)”. As illustrated in the **Response Document Figure #2** (presented as Fig. 3c in the revised manuscript), we designed forward primers priming at the upstream of DSB at the “Receiver” site to enrich the junction region with a downstream reverse Tn5 primer. By sequencing the amplicons, the “Donor” sites that were translocated to the “Receiver” site were identified. We found that only the off-targets assigned with a positive “translocation score” could serve as the “Donor” sites, which demonstrated the specificity of PEAC-seq when characterizing translocations.

Regarding inversion, PEAC-seq could infer an inversion event from two “relevant” translocation events on one chromosome. Specifically, if an identified translocation event fused the ends of two DSBs and another translocation event fused different ends of the same two DSBs, an inversion could be inferred. PEAC-seq could not directly identify “inversion” but need to combine with UDiTas-based assay to find out the “Donor” site of translocation.

Response Document Figure #3 (presented as Fig. 3c in the revised manuscript) An enrichment-based method, UDiTas, was used to identify Donor sites (red sequence on the right) from a PEAC-seq identified translocation. Two forward primers were designed to prime at the upstream of the DSB at the “Receiver” site to enrich the junction sequences at the translocation site by nested PCR, which was followed by Amplicon-seq.

The revised text was quoted as follow:

“Fig. 3 PEAC-seq identified DNA translocations relevant to CRISPR genome editing

The design of validation PCR to identify the genomic sequence of the Donor Sites. Two specific primers (Nest-F1 and Nest-F2) were designed upstream of the gRNA of the Receiver Site. The Nest-F1 and Nest-F2 were sequentially used with the downstream Tn5 primer, and two amplicons were generated. The 2nd amplicons were sent for Amplicon-seq.”

5) A major limitation of this study, like all previous studies describing new methods for CRISPR off-target detection, is that the Authors only provide a list of off-targets and compare the number of off-targets detected by different methods. Instead, it would be much more relevant to focus on where, along the genome, off-target events occur: is there a specific sequence or epigenome context that favours the formation of off-target alterations? The Authors should intersect their translocation data with a variety of available epigenome tracks (e.g., ATAC-seq or histone mark ChIP-seq data from the same or related cell lines) trying to understand whether there exist specific features that prime these regions to undergo unwanted editing. Even more importantly, the Authors should investigate if the observed translocations affect cancer-related genes and compare off-target translocation breakpoints with those of cancer translocations found in human cancers (e.g., using the TCGA gene fusion database).

Response: We thank the Reviewer for bringing this to our attention. As suggested, we have analyzed the genomic co-localizations between the PEAC-seq off-targets and epigenetic signals. We plotted the density of ATAC-seq peaks and ChIP-seq peaks of multiple histone modifications⁴ and proteins surrounding (± 5 kb) the PEAC-seq off-targets. Briefly, the results indicated that off-targets tended to occur in open chromatin regions (ATAC-seq) and to be associated with histone modifications in active gene regulation (H3K4me3, H3K9ac, and H3K27ac) and gene transcription (POLR2A) (**Response Document Figure #4**).

Response Document Figure #4 (presented as Fig. 6a in the revised manuscript) Epigenetic signals surrounding ± 5 kb of the PEAC-seq off-targets

However, PEAC-seq translocation does not associate with the above epigenetic marks as well as cancer-related fusion genes. We further examined the

co-occurrence of PEAC-seq translocations and double strand breaks (DSBs), which we inferred from the dsODN only control sample (no Cas9/gRNA) of GUIDE-seq conducted on VEGFA TS1, TS2, and TS3 in HEK293T cells. Compared to equally sized regions re-sampled randomly from across the genome, we observed significant enrichment of DSBs surrounding $\pm 5\text{kb}$ of the PEAC-seq translocation sites, indicating that CRISPR editing-induced translocation tends to occur at DSB enriched regions (**Response Document Figure #5**).

Response Document Figure #5 (presented as Fig. 6b in the revised manuscript) DSB signals around the $\pm 5\text{kb}$ of the PEAC-seq identified translocations and control regions. Left panel: DSB signals surrounding the translocation sites; Right panel: DSB signals surrounding the matched control sites.

We discussed the genomic context of off-targets and translocations in the revised manuscript as quoted below:

“Finally, it is intrinsically interesting that not all potential off-target sequences are eventually edited as off-targets. To look into this question, we analyzed the genomic co-localizations between the PEAC-seq off-targets and epigenetic signals collected from public data (ref). We plotted the density of ATAC-seq peaks and ChIP-seq peaks of multiple histone modifications and proteins surrounding ($\pm 5\text{kb}$) the PEAC-seq off-targets. Briefly, the results indicated that off-targets tended to occur in open chromatin regions (ATAC-seq) and to be associated with histone modifications in active gene regulation (H3K4me3, H3K9ac, and H3K27ac) and gene transcription (POLR2A, EP300, H2AFZ) (Fig. 6a). PEAC-seq translocation does not associate with the above epigenetic marks as well as cancer-related fusion genes, but show co-occurrence with the double strand breaks (DSBs) in HEK293T cells (Methods). Compared to control regions, which were equally sized regions re-sampled randomly across the genome, we observed enrichment of DSBs surrounding $\pm 5\text{kb}$ of the PEAC-seq translocation sites (Fig. 6b), indicating that CRISPR editing-induced translocation tends to occur at DSB enriched regions.”

ADDITIONAL REMARKS

1) Throughout the manuscript and in figures, it is never specified in which cell line(s) the experiments were conducted (HEK293T?). Please provide this important information.

Response: We apologize for missing this important information. As suggested, we have now included the cell line (HEK293T) in the Results section whenever needed. We also included the information in the Methods.

2) The Authors often use the expression *in cellulo* to describe experiments conducted in cultured cells. Although this expression is often encountered in scientific articles, it is grammatically incorrect because in Latin the word 'cell' (*cellula*) is female. Therefore, the correct expression is *in cellula*.

Response: We thank the Reviewer for this valuable suggestion. We have changed 'cellulo' to 'cellula' in the revised manuscript.

3) Figure panel labels should be in small letters (e.g., Fig. 1a, b, c) as per Nature Communications style.

Response: We thank the Reviewer for bringing this to our attention. We have modified the figure panel labels to small letters in the revised manuscript.

4) Fig. 1e: I assume that the schemes on the right indicate the strand and orientation of the PEAC-seq cassette (red). However, this is not explained and is different than what is shown in Fig. 1a, b where both strands of the cassette are colored in red. The Authors should clarify this, both in the figures and in the main text.

Response: As suggested, we have modified the PEAC-seq cassette in the Fig. 1e (Fig. 2c in the revised manuscript) and used the same colors as in the Fig. 1a and 1b. Explanation for the colors used in the illustration has been added in both the figure caption (**Response Document Figure #1**) and main text.

5) The Authors should consider providing supplementary BED files allowing the visualization of read pileups at on- and off-target sites, instead of showing all of them in Supplementary Figures that extend beyond one page.

Response: Thanks for the Reviewer's suggestion. We have uploaded BED files and TDF files to GEO (GSE179523 and GSE179436) for data visualization in genome browser.

References

-
- 1 Anzalone, A. V. *et al.* Search-and-replace genome editing without double-
strand breaks or donor DNA. *Nature* **576**, 149–157, doi:10.1038/s41586-019-
1711-4 (2019).
- 2 Veres, A. *et al.* Low incidence of off-target mutations in individual
CRISPR-Cas9 and TALEN targeted human stem cell clones detected by whole-
genome sequencing. *Cell Stem Cell* **15**, 27–30, doi:10.1016/j.stem.2014.04.020
(2014).
- 3 Giannoukos, G. *et al.* UDiTaS, a genome editing detection method for indels
and genome rearrangements. *BMC Genomics* **19**, 212, doi:10.1186/s12864-018-
4561-9 (2018).
- 4 Boix, C. A., James, B. T., Park, Y. P., Meuleman, W. & Kellis, M.
Regulatory genomic circuitry of human disease loci by integrative
epigenomics. *Nature* **590**, 300–307, doi:10.1038/s41586-020-03145-z (2021).
- 5 Tsai, S. Q. *et al.* GUIDE-seq enables genome-wide profiling of off-target
cleavage by CRISPR-Cas nucleases. *Nat Biotechnol* **33**, 187–197,
doi:10.1038/nbt.3117 (2015).
- 6 Nelson, J. W. *et al.* Engineered pegRNAs improve prime editing efficiency.
Nat Biotechnol **40**, 402–410, doi:10.1038/s41587-021-01039-7 (2022).
- 7 Chen, P. J. *et al.* Enhanced prime editing systems by manipulating cellular
determinants of editing outcomes. *Cell* **184**, 5635–5652 e5629,
doi:10.1016/j.cell.2021.09.018 (2021).
- 8 Zong, Y. *et al.* An engineered prime editor with enhanced editing efficiency
in plants. *Nat Biotechnol* **40**, 1394–1402, doi:10.1038/s41587-022-01254-w
(2022).
- 9 Chiarle, R. *et al.* Genome-wide translocation sequencing reveals mechanisms
of chromosome breaks and rearrangements in B cells. *Cell* **147**, 107–119,
doi:10.1016/j.cell.2011.07.049 (2011).
- 10 Hu, J. *et al.* Detecting DNA double-stranded breaks in mammalian genomes by
linear amplification-mediated high-throughput genome-wide translocation
sequencing. *Nat Protoc* **11**, 853–871, doi:10.1038/nprot.2016.043 (2016).
- 11 Yin, J. *et al.* Optimizing genome editing strategy by primer-extension-
mediated sequencing. *Cell Discov* **5**, 18, doi:10.1038/s41421-019-0088-8
(2019).

Response to Reviewer #2

General comments;

CRISPR-based genome editing tools have widely been used for precise gene knockout and correction. But, nucleases-based potential limitations such as off-target editing, large deletion or chromosomal rearrangements have been continuously raised. To identify genome-wide off-target sites is a steady important issue in the genome editing field. In the present study, Yu et al developed PEAC-seq by adopting the prime editing technology (with not Cas9 nickase but wtCas9) to detect Cas9-mediated off-target and DNA translocation sites. Basically the PEAC-seq might be another in vivo approach for off-target searching, but I would like to raise a critical issue on the efficacy of this tool as follows.

1. I am skeptical on that the wtCas9 based prime editing may not have editing activity as much as the wtCas9 nucleases. For example, when ten arbitrary sites are selected, wtCas9 typically shows editing (or indel) frequencies about 50%, while wtCas9 based prime editing might show much less efficiencies (less than ~ 10%), which potentially indicates that the PEAC-seq could underestimate the off-target sites. I would like to emphasize that underestimation of off-target sites would be much worse than overestimation of them, in terms of a safety issue.

Response: Thanks for focusing our attention on this. We have now conducted new experiments that directly addressed the potential differential editing efficiency of wtCas9 based prime editing vs. wtCas9 nucleases. Briefly, we selected ten sites that were examined in the GUIDE-seq paper⁵ and performed genome editing in HEK293T cells using 1) wtCas9-MMLV (*i.e.*, the PEAC-seq construct) or 2) wtCas9. Happily, and arguing against the idea of differential editing activity, the results indicate that adding the MMLV component to wtCas9 did not substantially disrupt the editing activity of wtCas9. That is, the editing frequencies were comparable for each site for the wtCas9-MMLV and wtCas9 cells, and there was no consistent trend of one editor outperforming the other. Thus, we now have direct evidence supporting that our PEAC-seq system, as designed, effectively created indels at the ten examined sites and does not introduce an additional bias at the editing efficacy level that would lead to underestimation of off-target sites (**Response Document Figure #6**).

Response Document Figure #6 Indel frequencies at ten sites upon CRISPR editing by wtCas9-MMLV and wtCas9.

1-1. It is necessary to prove that the PEAC-seq does not generally underestimate the off-target sites compared to previous tools (i.e., other in vivo based tools such as GUIDE-seq, Discover-seq, Site-seq) with at least 3 different gRNAs in the MAIN Figure. One gRNA shown in Figure 1C is not enough. I concern that the PEAC-seq might miss several bona-fide off-target sites as shown in the Supplementary Figure 6.

Response: We would first like to mention that there was no Amplicon-seq data for the VEGFA TS2, VEGFA TS3, EMX2, FANCF, and RNFs in the initial submission to verify the PEAC-seq-unique and GUIDE-seq-unique off-targets. To address this Reviewer's comment, we have now obtained new data from additional Amplicon-seq for the FANCF and EMX1 sites for all the PEAC-seq-unique and GUIDE-seq-unique off-targets. Specifically, in addition to the VEGFA TS1 site presented in Fig 1c of the originally submitted manuscript, we have properly obtained validated datasets for the FANCF and EMX1 sites (**Response Document Figure #7**). We have added these new data to the main figure (Fig. 1c-1e in the revised manuscript). Our new results for the FANCF and EMX1 sites are consistent with the trends detected for the VEGFA TS1 site. Amplicon-seq supported that all GUIDE-seq unique off-targets at the FANCF site did not occur in the sample that we performed the PEAC-seq, suggesting good sensitivity and specificity of PEAC-seq. Amplicon-seq also showed that two out of twelve GUIDE-seq unique off-targets at the EMX1 site might occur in the sample that we performed the PEAC-seq. However, these two sites were identified later by ePEAC-seq, an improved version of PEAC-seq (Fig. 5c in the revised manuscript). Thus, we now have data from comparisons between PEAC-seq and GUIDE-seq for three sites supporting that PEAC-seq does not generally underestimate the off-target sites compared to previous tools.

Response Document Figure #7 (presented as Fig. 1c-e in the revised manuscript) Comparison of PEAC-seq and GUIDE-seq off-targets at three sites

The revised text was quoted as follow:

“At the sites of VEGFA TS1, VEGFA TS2, and VEGFA TS3, a large proportion of PEAC-seq off-targets were also reported by GUIDE-seq, but both methods hold a few unique off-targets (Fig. 1c, Supplementary Fig. 4-5). At the sites of FANCF, EMX1, and RNF2, all PEAC-seq off-targets were reported by GUIDE-seq (Fig. 1d-1e, Supplementary Fig. 7). We then conducted Amplicon-seq to verify those off-targets that were only identified by GUIDE-seq or PEAC-seq at VEGFA TS1, FANCF, and EMX1 sites⁵ (Supplementary Materials 1). At the VEGFA TS1 site, Amplicon-seq confirmed the two PEAC-seq-unique off-targets, demonstrating good sensitivity of PEAC-seq. For the GUIDE-seq-unique off-targets, all six off-targets at the FANCF site were confirmed not to occur in our sample, while two out of the twelve GUIDE-seq-unique off-targets at the EMX1 site and two out of the eight GUIDE-seq-unique off-targets at the VEGFA TS1 site were detected by Amplicon-seq. These data argued that PEAC-seq could effectively and specifically identify off-targets with a streamlined procedure without incorporating other exogenous reagents to tag and enrich these

sites.”

“We modified the PEAC-seq by using epegRNA (engineered pegRNA, incorporated 3' RNA structural motif evopreQ1)⁶ and including transient expression of MLH1dn⁷ with Cas9-MMLV. We did not include the truncated MMLV, as it is reported to be effective in plants but not in mammal cells⁸. By incorporating epegRNA, hMLH1, and epegRNA plus MLH1dn, we developed three modified versions of PEAC-seq and benchmarked their performances on identifying off-targets at EMX1 and VEGFA TS2 sites (Fig. 5a). We specifically concentrated to the PEAC-seq tag insertion, whose efficiency is critical to the overall performance of PEAC-seq. Among all modifications, incorporating epegRNA appears to be the most effective modification to increase the number of PEAC-seq tag insertion at different cutoffs (Fig. 5b). We named the epegRNA version of PEAC-seq as ePEAC-seq. Importantly, ePEAC-seq successfully identified the two missed off-target of EMX1 (Fig. 5c-d), emphasized its higher sensitivity than PEAC-seq. At the VEGFA TS2 site, ePEAC-seq also called more off-target sites shared with GUIDE-seq, comparing to PEAC-seq (Supplementary Fig. 4a & 11).”

2. The incorporate tag sequences by the PEAC-seq would be very critical but the information of it is not mentioned in the main text. What is the length of it? It is well known that the editing efficiency is negatively correlated to the length of the insertion sequences. Hence, when the incorporate tag is about 20-bp length, the precise incorporation efficiency might be very low even at the on-target site.

If then, the precise incorporation efficiency at off-target sites would be extremely low, indicating that the PEAC-seq is not a sufficient tool for searching genome-wide off-targets.

Response: As suggested, we have now conducted Amplicon-Seq to quantify the efficiency of tag insertion at ten sites from the GUIDE-seq paper⁵. The results demonstrated that the insertion efficiencies of the full-length tag were 11%-31% (**Response Document Figure #8**). It is reasonable to expect that the actual tag insertion efficiency is higher than these numbers, as a partial insertion could also generate amplicons to enrich the insertion tags from gRNA.

Response Document Figure #8 Efficiency of tag insertion at ten sites

Additionally, by introducing ePEAC-seq, an improved version of PEAC-seq, we demonstrated that the insertion efficiency could be further improved (**Response Document Figure #9**).

Response Document Figure #9 (presented as Fig. 5b in the revised manuscript) The improved PEAC-seq design of “epegQ1-Cas9-MMLV” further increased the length of inserted PEAC-seq tag

We used a 21-bp insertion tag sequence in the PEAC-seq construct (presented as Table 9 in the revised manuscript). The sequence should provide sufficient length for priming the enrichment primer while not too long to impact the insertion efficiency. RNA secondary structure and sequence uniqueness to the host genome have also been considered.

We have added two paragraphs in the revised text regarding to this matter:

“We designed a 21-nt insertion tag, with the consideration of (1) avoiding the RNA secondary structure of the insertion tag and between the insertion tag and the gRNA scaffold, (2) sequence uniqueness in the host genome, (3) sufficiently long for effectient anneal by PCR primers for enrichment.”

“Further, the insertion efficiency of PEAC-seq could also be related

to the length and sequence composition of the insertion tag (sTable 13). The RNA secondary structure of the insertion tag and sequence uniqueness to the host genome could vary across different pegRNA. However, this sequence is exchangeable as long as the above rules were considered, and we also provided a few other tested sequences (sTable 14).”

3. *Prime editing efficacies vary according to the RTT and PBS lengths. The strategy to design an optimized pegRNA for the PEAC-seq should be suggested.*

Response: Thanks for this Reviewer’s suggestion.

Regarding the PBS lengths, we tested the 13-nt template and 17-nt template, referring to the native PE paper. Both lengths worked equally well in our hands, and we used 13-nt as suggested by the PE paper¹.

Regarding the RTT lengths, we considered that the sequence should be sufficiently long to provide effective annealing in PCR enrichment but not too long to impact the insertion efficiency of the PEAC-seq tag. We also included our considerations when designing the PBS sequences in the Results and provided a few optional sequences in the Discussion (as quoted below).

Additionally, the Prime Editor system has been modified in many facets to improving editing efficiencies since it was published. We incorporated epegRNA (engineered pegRNA, incorporated 3’ RNA structural motif evopreQ1)⁶ and transient expression of MLH1dn⁷ with Cas9-MMLV to the PEAC-seq system (**Response Document Figure #10**) and examined the insertion efficiencies in the on-target site of VEGFA TS2. The results suggested a superlative insertion performance when incorporating the epegRNA into the PEAC-seq design (**Response Document Figure #9**), which we proposed as an improved version of PEAC-seq and named ePEAC-seq. We quoted below, we summarized these contents and added a new paragraph and figure in the revised manuscript.

Response Document Figure # 10 (presented as Fig. 5a in the revised manuscript) Schematic representation of the three modified versions of PEAC-seq.

The related contents in the revised manuscript were quoted as below:

“We designed a 21-nt insertion tag, with the consideration of (1) avoiding the RNA secondary structure of the insertion tag and between the insertion tag and the gRNA scaffold; (2) sequence uniqueness to the host genome; (3) sufficiently long for efficient anneal by PCR primers for enrichment.”

“ePEAC-seq, an improved version of PEAC-seq utilizing epegRNA

Since the original PEAC-seq protocol has been developed, multiple strategies have been proposed to improve the editing efficiency of the native PE system, including modifications on pegRNA⁶, MMLV⁸, and transient expression of a dominant negative MMR (DNA mismatch repair) protein⁷. We modified the PEAC-seq by using epegRNA (engineered pegRNA, incorporated 3' RNA structural motif evopreQ1) and including transient expression of MLH1dn with Cas9-MMLV. We did not include the truncated MMLV, as it is reported to be effective in plants but not in mammal cells⁸. By incorporating epegRNA, hMLH1, and epegRNA plus MLH1dn, we developed three modified versions of PEAC-seq and benchmarked their performances on identifying off-targets at EMX1 and VEGFA TS2 sites (Fig. 5a). We specifically concentrated to the PEAC-seq tag insertion, whose efficiency is critical to the overall performance of PEAC-seq. Among all modifications, incorporating epegRNA appears to be the most effective modification to increase the

number of PEAC-seq tag insertion at different cutoffs (Fig. 5b). We named the epegRNA version of PEAC-seq as ePEAC-seq. Importantly, ePEAC-seq successfully identified the two missed off-target of EMX1 (Fig. 5c-d), emphasized its higher sensitivity than PEAC-seq. At the VEGFA TS2 site, ePEAC-seq also called more off-target sites shared with GUIDE-seq, comparing to PEAC-seq (Supplementary Fig. 4a & 11).”

“Further, the insertion efficiency of PEAC-seq could also be related to the length and sequence composition of the insertion tag (sTable 13). The RNA secondary structure of the insertion tag and sequence uniqueness to the host genome could vary across different pegRNA. However, this sequence is exchangeable as long as the above rules were considered, and we also provided a few other tested sequences (sTable 14). Another improvement may include a few random nucleotides in the PBS region of pegRNA to achieve higher extension efficiency at off-targets with mismatched nucleotides in PBS. In terms of the length of the PBS (primer binding site), we inherited a 13-nt design according to the native PE system¹, although both the 13-nt and 17-nt worked equally well in our hands.”

(minor)

- In the title, ‘adapt’ is correct? ‘adopt’ might be possible.

Response: Thanks for this suggestion. We have changed the ‘adapt’ to ‘adopt’ in the title and in the text.

- In page 4, ‘Supplementary FigX’ should be corrected.

Response: Thanks for bringing this omission to our attention. We now refer the correct Supplementary Figures across the manuscript.

- It is not easy to understand the meaning of the Figure 2C. Do inserted positions vary within gRNA?

Response: We apologize for the confusion. To clarify, the inserted positions are consistent, and we have updated the figure caption and relevant text.

“Fig. 2 Analysis on the PEAC-seq off-target sites

Mutation frequencies were plotted at each position alongside the gRNA and PAM sequences (from 5’ to 3’). From top to bottom are profiles of VEGFA TS1, TS2 and TS3.”

“Furthermore, we also examined whether the position of

mismatches on the pegRNA sequence might affect the off-target identification⁵, especially in the primer binding site (PBS) that is crucial to initiate the primer extension of reverse transcription¹⁸. To do that, we grouped the off-target sequences from the “Shared”, “PEAC-seq-unique”, and “GUIDE-seq-unique” and aligned with the on-target sequence and PAM sequences. The mutation frequency at each position were plotted for three sites (Fig. 2e). The patterns among the shared and unique off-target groups were quite consistent in VEGFA TS2 (81 sites) and VEGFA TS3 (35 sites), but a bit fluctuated in VEGFA TS1 (24 sites). Although the smaller number of off-targets of VEGFA TS1 might contribute to its fluctuated mutation frequency, this result indicated that the sensitivity of PEAC-seq might be affected by mismatches located in the PBS region of PEAC-seq. Actually, the two verified GUIDE-seq-unique off-targets of TS1 both show mismatches in PBS region (at the position 14 and 17 of the spacer, respectively) (Table S12). Nevertheless, off-target identification of the TS3 gRNA seems more tolerant to PBS mutations, which implied that the extent of the influence might be site-specific.”

- *In the discussion section, the authors argued that the PEAC-seq is the only tool for detecting translocation sites, but HTGTS is a dedicated tool for detecting CRISPR-mediated translocation which was already published in 2015 [PMID: 25503383].*

Response: We apologize for the confusion with our previous statement. The HTGTS is designed to detect CRISPR-mediated translocation when one side of the translocated sequences is known. And PEAC-seq identified translocations based on abnormal sequencing reads without prior information from either side of sequences. We should have properly defined the scope of each method.

We have revised the language as follows:

“Methods have been developed to identify DNA translocations, but the sequence information of at least one end of the rearranged DNAs is usually required, e.g., HTGTS^{3,9-11}. And a systematic identification of DNA translocation is still lacking.”

References

- 1 Anzalone, A. V. *et al.* Search-and-replace genome editing without double-strand breaks or donor DNA. *Nature* **576**, 149–157, doi:10.1038/s41586-019-1711-4 (2019).
- 2 Veres, A. *et al.* Low incidence of off-target mutations in individual CRISPR-Cas9 and TALEN targeted human stem cell clones detected by whole-genome sequencing. *Cell Stem Cell* **15**, 27–30, doi:10.1016/j.stem.2014.04.020 (2014).

-
- 3 Giannoukos, G. *et al.* UDiTaS, a genome editing detection method for indels
and genome rearrangements. *BMC Genomics* **19**, 212, doi:10.1186/s12864-018-
4561-9 (2018).
- 4 Boix, C. A., James, B. T., Park, Y. P., Meuleman, W. & Kellis, M.
Regulatory genomic circuitry of human disease loci by integrative
epigenomics. *Nature* **590**, 300–307, doi:10.1038/s41586-020-03145-z (2021).
- 5 Tsai, S. Q. *et al.* GUIDE-seq enables genome-wide profiling of off-target
cleavage by CRISPR-Cas nucleases. *Nat Biotechnol* **33**, 187–197,
doi:10.1038/nbt.3117 (2015).
- 6 Nelson, J. W. *et al.* Engineered pegRNAs improve prime editing efficiency.
Nat Biotechnol **40**, 402–410, doi:10.1038/s41587-021-01039-7 (2022).
- 7 Chen, P. J. *et al.* Enhanced prime editing systems by manipulating cellular
determinants of editing outcomes. *Cell* **184**, 5635–5652 e5629,
doi:10.1016/j.cell.2021.09.018 (2021).
- 8 Zong, Y. *et al.* An engineered prime editor with enhanced editing efficiency
in plants. *Nat Biotechnol* **40**, 1394–1402, doi:10.1038/s41587-022-01254-w
(2022).
- 9 Chiarle, R. *et al.* Genome-wide translocation sequencing reveals mechanisms
of chromosome breaks and rearrangements in B cells. *Cell* **147**, 107–119,
doi:10.1016/j.cell.2011.07.049 (2011).
- 10 Hu, J. *et al.* Detecting DNA double-stranded breaks in mammalian genomes by
linear amplification-mediated high-throughput genome-wide translocation
sequencing. *Nat Protoc* **11**, 853–871, doi:10.1038/nprot.2016.043 (2016).
- 11 Yin, J. *et al.* Optimizing genome editing strategy by primer-extension-
mediated sequencing. *Cell Discov* **5**, 18, doi:10.1038/s41421-019-0088-8
(2019).

Reviewers' Comments:

Reviewer #1:

Remarks to the Author:

The Authors have satisfactorily addressed all my questions and performed the extra experiments that I had suggested. In addition, the clarity of the main text and figures is now substantially improved. Therefore, I am now happy to recommend this manuscript for publication in Nature Communications.

Reviewer #2:

Remarks to the Author:

The authors have mostly answered the issues I raised in the earlier review. However, in this 2nd round, I would like to further know the opinions from the authors. I think that the PEAC-seq technique is relevant to detect CRISPR-mediated DNA translocation sites, but is less suitable to detect the genome-wide off-target sites, compared to other tools such as GUIDE-seq. In the prime editing system, the pegRNA consists of the primer binding site (PBS) and RT template (i.e., "inserting tag" in Fig.1a in this study). Because the PBS sequences are obtained from the on-target site, I assume that, in case of off-target sites in other locus, the PBS would be much different, which might decrease the prime editing efficiencies in the off-target sites and result in the underestimate the off-target sites as shown in the Figure 1, Supplementary Fig 4 and 5. It should be discussed further.

Response to Reviewer #2

The authors have mostly answered the issues I raised in the earlier review. However, in this 2nd round, I would like to further know the opinions from the authors. I think that the PEAC-seq technique is relevant to detect CRISPR-mediated DNA translocation sites, but is less suitable to detect the genome-wide off-target sites, compared to other tools such as GUIDE-seq. In the prime editing system, the pegRNA consists of the primer binding site (PBS) and RT template (i.e., “inserting tag” in Fig.1a in this study). Because the PBS sequences are obtained from the on-target site, I assume that, in case of off-target sites in other locus, the PBS would be much different, which might decrease the prime editing efficiencies in the off-target sites and result in the underestimate the off-target sites as shown in the Figure 1, Supplementary Fig 4 and 5. It should be discussed further.

Response: We thank that Reviewer #2 further asked our opinions on the PBS mismatches and their influences on the prime editing efficiencies at off-target sites. We agreed with this reviewer that PBS mismatches appear to affect the sensitivity of PEAC-seq and are happy to discuss it further and propose possible improvements.

First, although systematic off-target analysis indicated that mismatches located between the 14-17 nucleotides (5' to 3') at the putative off-targets are typically less tolerant by Cas9 and variants¹, we noticed that the two missing off-targets of VEGFA TS1 (GUIDE-seq-unique off-targets) did include PBS mismatches adjacent to the starting point of primer extension of reverse transcription (**Response document figure #1**). Similarly, in the GUIDE-seq-unique off-targets at the VEGFA TS2 and VEGFA TS3 sites, not verified by Amplicon-seq in our samples though, we observed relatively more PBS mismatches compared to the shared and PEAC-seq-unique off-targets (**Response document figure #2-3**). However, many off-targets with PBS mismatches were also successfully identified by PEAC-seq, indicating the complication of the effects of PBS mismatches on reverse transcription (**Response document figure #1-3**). Second, commercial reverse transcriptase (e.g., superscript IV) is typically applied at 50-55 °C when using oligo d(T)₂₀ or gene-specific primer and at 23 °C when using random hexamer. We thus reasoned that 13-nt PBS should allow effective reverse transcription at 37 °C in most cases.

Response Document Figure #1 (presented as sFig. 12a in the revised manuscript) PBS mismatches in *VEGFA* TS1 off-targets. GUIDE-seq unique and PEAC-seq unique off-targets were verified by Amplicon-seq.

Response Document Figure #2 (presented as sFig. 12b in the revised manuscript) PBS mismatches in *VEGFA* TS2 off-targets. PEAC-seq off-targets were compared to those identified by GUIDE-seq but without Amplicon-seq validation.

VEGFA TS3

Response Document Figure #3 (presented as sFig. 12c in the revised manuscript) PBS mismatches in *VEGFA* TS3 off-targets. PEAC-seq off-targets were compared to those identified by GUIDE-seq but without Amplicon-seq validation.

Nevertheless, we propose to include a few random nucleotides in the PBS regions of pegRNA (mut-pegRNA), especially proximal to the primer extension site, to improve the extension efficiency at off-targets with PBS mismatches (**Response document figure #4**). According to this study's PEAC-seq and ePEAC-seq data, pegRNA designed from the on-target sequence could enable PEAC-seq tag insertion in most off-target sites, and the incorporation of mut-pegRNA may improve the insertion efficiency of PEAC-seq tags in some off-target sites with critical PBS mismatches.

Response Document Figure #4 (presented as sFig. 13 in the revised manuscript) Mix pegRNA and mut-pegRNAs may increase the insertion efficiency of the PEAC-seq tag, for example, include 50% pegRNA and 50% mut-pegRNAs with 10% of each.

Besides incorporating mut-pegRNA, reverse transcriptase evolving for error-correcting activity may further improve the primer extension efficiencies, *e.g.*, proofreading reverse transcriptase². The currently used MMLV in Prime Editor, PEAC-seq, and ePEAC-seq has no 3' to 5' exonuclease activity. If proper enzyme could be evolved and characterized, the 3' to 5' exonuclease activity could correct mismatches between PBS and off-targets and may further increase the primer extension efficiency.

We have included a summary of this contents in the Discussion section in the revised manuscript, as quoted below (words in *italic* were updates from this version of revised manuscript).

“The limitation of this study, however, is that the insertion efficiency of the PEAC-seq tag might vary across different pegRNAs and at different off-targets. For each pegRNA, the RNA secondary structure of the insertion tag and sequence uniqueness to the host genome could vary. But if the aforementioned guidelines were taken into account, this sequence is interchangeable, and we have supplied a few additional tested sequences (sTable 13-14). Regarding the PBS (primer binding site) length, we inherited a 13-nt design according to the native PE system³, although both the 13-nt and 17-nt worked equally well in our hands. And the PBS sequences, which were derived from the on-target sites, can be different at off-target sites. Mismatches between the PBS and the spacer

sequences at off-target sites might affect primer extension in the reverse transcription and result in low insertion efficiencies of the PEAC-seq tag. *Actually, the two missing off-targets in the VEGFA TS1 site include PBS mismatches at positions 14 and 17 (5' to 3') at the off-target sites (Fig. 1c, sTable 12), which are proximal to the starting point of primer extension of reverse transcription (Supplementary Fig. 12a). GUIDE-seq-unique off-targets in the VEGFA TS2 and VEGFA TS3, not verified by Amplicon-seq though, also contained relatively more PBS mismatches compared to the shared and the PEAC-seq-unique off-targets (Supplementary Fig. 12b-c). However, many off-targets with PBS mismatches were successfully identified by PEAC-seq, indicating the complication of the effects of PBS mismatches on reverse transcription. Nevertheless, we propose to include a few random nucleotides in the PBS regions of pegRNA (mut-pegRNA) (e.g., proximal to the primer extension site) to improve the extension efficiency at off-targets with PBS mismatches (Supplementary Fig. 13). According to this study's PEAC-seq and ePEAC-seq data, pegRNA designed from the on-target sequence could enable PEAC-seq tag insertion in most off-target sites, and the incorporation of mut-pegRNA might improve the insertion efficiency of PEAC-seq tags in some off-target sites with critical PBS mismatches. Besides, reverse transcriptase evolving for error-correcting activity (e.g., error-correcting reverse transcriptase²) may further improve the primer extension efficiencies. If proper enzyme could be evolved and characterized, the 3' to 5' exonuclease activity could correct mismatches between PBS and off-targets."*

1. Kim, N. et al. Prediction of the sequence-specific cleavage activity of Cas9 variants. *Nat Biotechnol* **38**, 1328-1336 (2020).
2. Ellefson, J.W. et al. Synthetic evolutionary origin of a proofreading reverse transcriptase. *Science* **352**, 1590-1593 (2016).
3. Anzalone, A.V. et al. Search-and-replace genome editing without double-strand breaks or donor DNA. *Nature* **576**, 149-157 (2019).

Reviewers' Comments:

Reviewer #2:

Remarks to the Author:

I appreciate the further efforts to address my additional concern and I agree to publish the revised manuscript in this Nature Communications journal.